# Shelter in Smoleń III – A unique example of stratified Holocene clastic cave sediments in Central Europe, a lithostratigraphic stratotype and a record of regional paleoecology

**Maciej T. Krajcarz** [1]*, **Marcin Szymanek**[2], **Magdalena Krajcarz**[3], **Andrea Pereswiet-Soltan**[4], **Witold P. Alexandrowicz**[5], **Magdalena Sudoł-Procyk**[3]

**1** Institute of Geological Sciences, Polish Academy of Sciences, Warszawa, Poland, **2** Faculty of Geology, University of Warsaw, Warszawa, Poland, **3** Institute of Archaeology, Nicolaus Copernicus University in Toruń, Toruń, Poland, **4** Department of Paleozoology, Institute of Environmental Biology, University of Wrocław, Wrocław, Poland, **5** Chair of General Geology and Geotourism, Faculty of Geology, Geophysics and Environment Protection, AGH University of Science and Technology, Kraków, Poland

* mkrajcarz@twarda.pan.pl

## Abstract

A cave site Shelter in Smoleń III (southern Poland) contains an approximately 2-m-thick stratified sequence of Upper Pleistocene and Holocene clastic sediments, unique for Central Europe. The sequence contents abundant fossil fauna, including mollusk, rodent and bat remains. The cave sites with long profiles of subfossil fauna present a great value for reconstructions of regional terrestrial paleoenvironment. We explore the stratigraphy of this site through analyses of the lithology and geochemistry of sediments, radiocarbon dating of faunal and human remains and charcoals, and archaeological study, as well as the paleoecology derived from the taxonomic composition of fossil faunal assemblages. Our data show that the entire period of the Holocene is recorded in the rockshelter, which makes that site an exceptional and highly valuable case. We present paleoenvironmental reconstructions of regional importance, and we propose to regard Shelter in Smoleń III as a regional stratigraphic stratotype of Holocene clastic cave sediments.

## Introduction

Cave sediments are important source of information for understanding the history of terrestrial geoecosystems (see, e.g., [1–5]). Due to their concave morphology and solid walls, many caves serve as sediment traps with the ability to continuously collect sediments for millennia [6]. Preserved sequences record long time intervals and are able to survive destructive geomorphological events, such as ice sheet advance, floods or deflation. Although the stratigraphic resolution of clastic and/or biogenic cave sediments is usually much lower than in subaqueous deposits, such as lacustrine or peat bog sediments, caves are among the rare landforms that record subaerial terrestrial conditions. In particular, data on past terrestrial fauna are well protected in caves (see, e.g., [7,8]). This outcome is not only because caves serve as shelters for

**Data Availability Statement:** All relevant data are within the paper and its Supporting Information files

**Funding:** This study was supported by the National Science Centre, Poland, grant Nos.: 2011/01/N/ HS3/01299 (awarded to MS-P; excavation works, IRSL dating, radiocarbon dating of charcoals, analysis of rodents and the salaries for MS-P, MK and MTK), 2014/13/D/HS3/03842 (awarded to MTK; radiocarbon dating of bones and the salaries for MK and MTK), 2017/27/B/NZ8/00728 (awarded to MK; radiocarbon dating of bones and salary for MK) and 2016/20/S/ST10/00163 (awarded to AP-S; analysis of chiropterans and the salary for AP-S); by the Institute of Geological Sciences of the Polish Academy of Sciences: internal grant "Jaskinie" (awarded to MTK; geological analyses) and the internal fund for young scientists 2013 (awarded to MK; radiocarbon dating of mollusks); and by the Faculty of Geology, University of Warsaw, internal grant No. BST 166901 (awarded to MS; analysis of mollusks). The funders had no role in study design, data collection and analysis, decision to publish, or preparation of the manuscript.

**Competing interests:** The authors have declared that no competing interests exist.

animals and are natural places of deposition of animal remains but also because a calcareous environment favors preservation of fossil bones and shells.

Caves have also been used by humans in prehistoric, historic and recent times as living places, temporary shelters, storerooms or sacral objects (e.g., [9,10]). Therefore, they are important archaeological sites, often with multiple cultural levels preserved in superposition and delivering data on the local or regional development of human beings and their culture. For this reason, it is crucial not only to recognize the stratigraphy of cave sediments in particular sites but also to detect regional lithological variability and similarities between sites. This would help to formulate more general stratigraphic schemes that can be used for further inter- and intraregional correlations.

Although numerous karst sites with abundant faunal remains have been reported from Central Europe, few of these sequences are stratified and go back as far as the final part of the Last Glaciation (e.g., [11–17]). Accumulations of over a meter thick packets of Late Glacial and Holocene clastic cave deposits are rather unique in the region and occur only at single sites, e.g., Veľká Drienčanská Cave in Slovakia[16], Barová Cave in Czech Republic[14] and Petény Cave in Hungary[17]. At the northern side of Carpathian Mountains, most of cave sites studied until now are situated in Kraków-Częstochowa Upland in southern Poland[18]. This is a karstic region with a very large number of caves and rockshelters[19]. Most of these features contain clastic sediments, in some cases studied by archaeologists, geologists and paleontologists. The lithology and stratigraphy of Pleistocene sediments were recognized in a number of sites in the region[20–28]. In opposition to the Pleistocene series, the Holocene sediments were poorly studied. The main reason is the focus of the scientific community on the Pleistocene and Paleolithic issues. An additional reason is that the internal stratification of the Holocene series in caves of the region is usually poor or completely lacking, and at the most sites, the entire Holocene sequence is limited to only one stratum (e.g., [20–23,28–31]). Most of these sequences are only a few-dozens-of-centimetres thick and represent only the Middle to Upper Holocene, or just the uppermost Upper Holocene[32–40]. This stratum has usually been described as a dark loam or humus layer. Explanation of this faint stratification is beyond the scope of this paper; however, the most likely reason is past human activity, which was especially intensive during the Medieval and post-Medieval periods[9,10,28]. To date, only a few sites with complex Holocene stratigraphy have been discovered in Kraków-Częstochowa Upland: Cave above the Słupska Gate[41]; Żytnia Skała Rockshelters[42]; Nad Mosurem Starym Dużą Cave[43]; Rockshelter in Zalas[32]; Zawalona Cave[13]; Żarska Cave, which was excavated recently[44]; most likely Zegar Cave[45]; Perspektywiczna Cave, which is still under excavation[27,46]; and Shelter in Smoleń III, which is the subject of this paper.

Despite this knowledge, there are still many uncertainties in our understanding of the stratigraphy of Holocene cave sediments in the region and their usefulness for paleoenvironmental reconstructions. The most important reason is a lack of solid chronological frameworks for most of the studied sites. Their chronostratigraphy is usually based on singular radiocarbon dates (e.g., [32,41]) and/or archaeological dating (e.g., [13,42–45]), which in many cases provides rather wide chronological windows. Another factor is lack of Lower Holocene or Lower-to-Middle Holocene at many sites. In some cases there is a hiatus between Upper Pleistocene and Middle or Upper Holocene (e.g., [32,42,43]), in others the entire sequence is restricted to the upper part of Holocene (e.g., [20–23,28–31]). With only several long sequences in hand we cannot be sure which of them are representative for the region, and which reflect only the local conditions. Therefore, new data are awaited to build a comprehensive reconstruction of paleoenvironment in the region, its changes through time and its spatial variability. Moreover, even the longest and the most complete sequences are usually biased by different degree of preservation and/or accumulation of fossil fauna. Due to this, the direct intra-site correlation

between independent paleoenvironmental proxies (such as fauna of rodents and malacofauna) are usually impossible or constrained.

Shelter in Smoleń III bears a complex stratified series of Late Glacial and Holocene cave sediments, which was found during an archaeological excavation from 2012–2013. Rich fossil material of snails and vertebrates (rodents and bats) preserved at this site allows reconstructing the paleoecological and paleoclimatic conditions for the recorded span of time. The aim of this study is to explore this rich material and to use the geological and paleontological data from Shelter in Smoleń III as the proxies to build a comprehensive litho-bioclimato-stratigraphic scheme, which can serve as a regional stratotype.

## The studied site

Shelter in Smoleń III (ShSIII) is situated near Smoleń village (Pilica comm., Zawiercie dist., southern Poland), in the central part of Kraków-Częstochowa Upland (Fig 1), in the microregion called Ryczów Upland[47]. The rockshelter is situated on the right side of the currently dry Wodąca Valley, 445 m a.s.l., E 19°40′27″, N 50°26′02″. Two entrances are situated next to one another; they are exposed to the NE and open to a small plateau in front of the rockshelter (Fig 2). The rockshelter is most likely a remnant of a larger collapsed cave system. Its total length is 8.8 m, including the branches[48], and the surface of sedimentary fill is approximately 11 m$^2$. The entire interior is controlled by external weather conditions and exposed to daylight, except for the rearmost area.

The rockshelter was named *Schronisko w Smoleniu III* (Shelter in Smoleń III) and initially listed under the inventory number 392[49], then IV.C.12[19], and finally J.Cz.IV-04.44[48]. The last number and the name are listed in the governmental database of the caves of Poland (http://jaskiniepolski.pgi.gov.pl/).

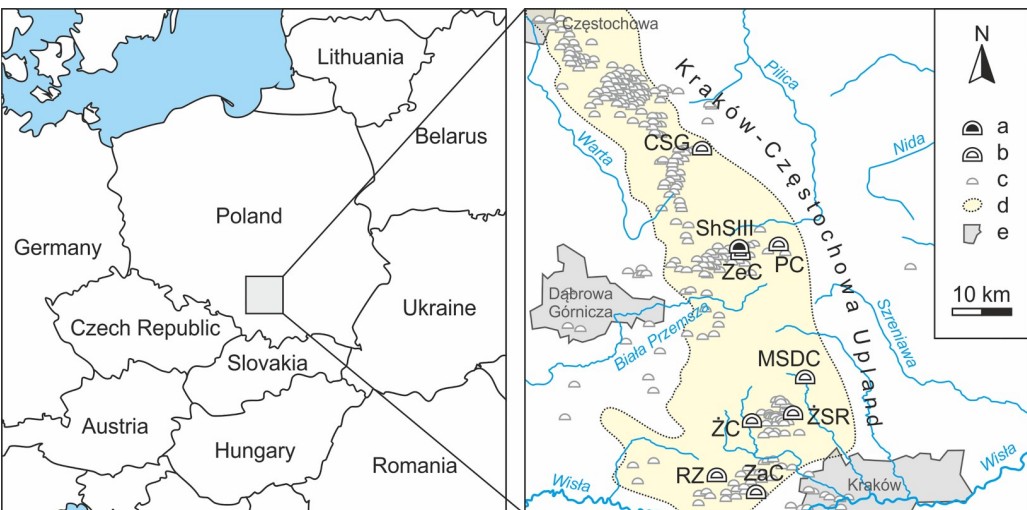

**Fig 1. Localization of ShSIII.** Its situation in Central Europe is shown in the left panel, and within Kraków-Częstochowa Upland in the right panel. Legend: a–ShSIII; b–other important cave sites of Holocene sediments and fossil fauna in the region (CSG–Cave above the Słupska Gate, MSDC–Nad Mosurem Starym Duża Cave, PC–Perspektywiczna Cave, RZ–Rockshelter in Zalas, ZaC–Zawalona Cave, ZeC–Zegar Cave, ŻC–Żarska Cave, ŻSR–Żytnia Skała Rockshelters); c–other caves and rockshelters (after the Polish governmental database http://www.jaskiniepolski.pgi.gov.pl/); d–macroregion of Kraków-Częstochowa Upland (according to [47]); e–the largest cities.

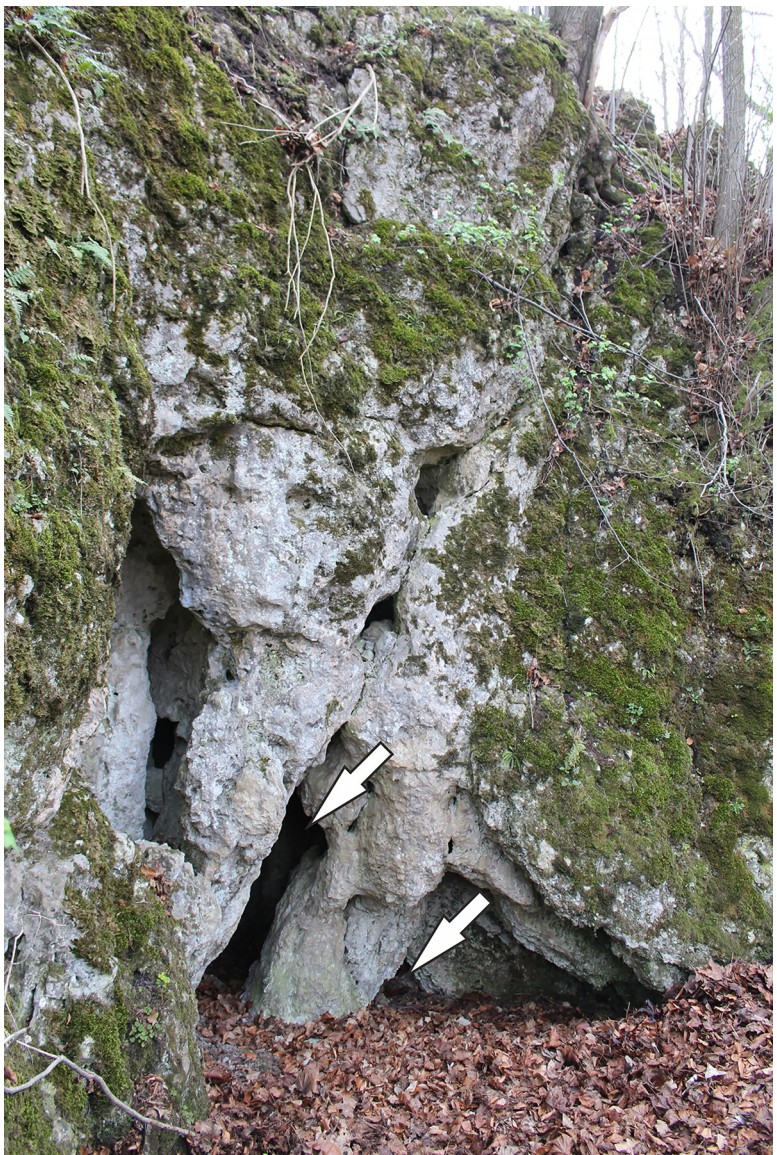

**Fig 2. The entrances of ShSIII.** A view from north-east, from the small plateau in front of the rockshelter. Two entrances are indicated with the arrows. Photo made in April 2014 by MS-P.

## Geological setting

The region is characterized by a variable topography with steep high limestone tors and cliffs and deeply cut valleys. The core of the microregion are the Upper Jurassic massive limestones, partially covered by red desert sands and loams dated to Paleogene/Neogene[50] and sandy deposits attributed to the fluvio-periglacial facies of the Middle Pleistocene[51,52]. Near the site, the sands are covered by loess deposits, which become thicker and tend to form a continuous coverage toward the east[53]. The karstic cavities (caves and rockshelters) in the microregion are filled with the Upper Pleistocene and Holocene sediments[23,45,54,55]. The most complex stratigraphic sequence has been found in the nearby Biśnik Cave, situated 1.1 km to southwest, where pre-Pleistocene, Middle-to-Upper Pleistocene and Holocene sediments have been preserved[26,56]. However, the Holocene sediments of Biśnik Cave exhibit reduced thickness and weak lithological variability[26].

## Archaeological site

The rockshelter was recognized as an archaeological site in 2012, during the research project focused on the ancient settlement pattern in the region[57]. The test pitting started in 2012, the excavation continued in 2013 and revealed a multi-episode sequence of human activity [58–60]. It is noteworthy that only minor attention has been given to the Holocene archaeological sites in caves of Kraków-Częstochowa Upland until last years[9], especially to the stratigraphic situation of the archaeological finds. Therefore, the thorough investigation of ShSIII, including the modern excavation standards, makes this site an important record of the Holocene human activity in the region.

## Material and methods

### Excavation and sampling

Permission for archaeological excavation was granted by the Regional Heritage Office (Polish: Wojewódzki Urząd Ochrony Zabytków w Katowicach; document No: K-AR.5161.61.2012.JP, ID: 3110, permission No 74/2012), and permission for the field works was granted by the Polish National Forests, regional office in Olkusz (Polish: Lasy Państwowe–Nadleśnictwo Olkusz) who is a land owner and forest manager (permission No: ZG5/503-1/2012 and ZG5/503-1/ 2013). The site was excavated from 2012–2013 according to the archaeological standards of the test pitting. Excavation covered 10 squares of the archaeological grid (approximately 7 m$^2$ of surface in total, Fig 3) and reached the bedrock, revealing a maximum 2.6-m-thick sequence of clastic sediments[57]. The deposits were divided into archaeological layers according to differences in the color, consistency and amount of limestone debris observed in the field. These layers well reflect the geological stratification, and in some cases, they represent soil horizons. The excavation was performed by removing 10-cm-thick intervals of sediment (or 5-cm-thick where the concentration of artifacts or fossils was high) from the squares. The arrangement of layers was recorded by drawing and photographing for each interval. For 3D recording of our work, we used a north-oriented meter grid for horizontal location (Fig 3) and depth below the

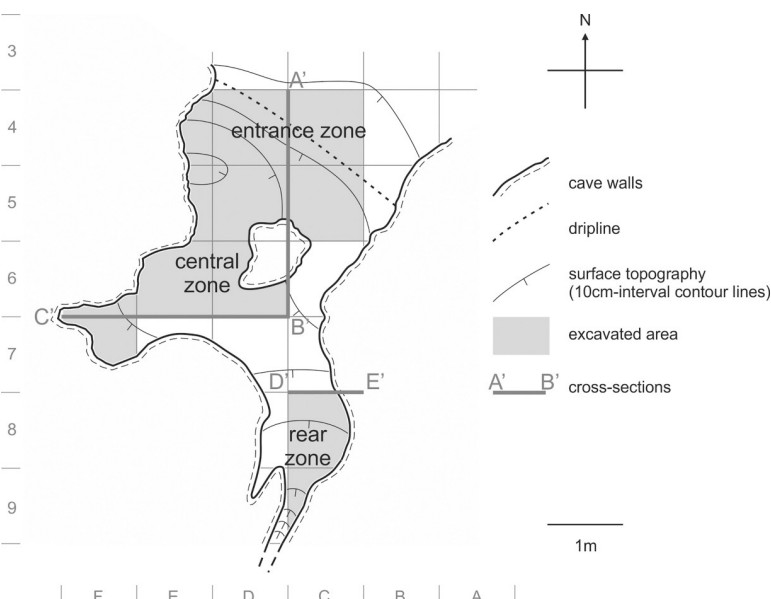

**Fig 3. Plan of ShSIII.** The cross-sections A'-B', B'-C' and D'-E' are shown in Fig 4. Drawn by MS-P and MTK.

site stratum (an arbitrarily selected point on the wall) for vertical location. The following finds and samples were collected:

- Archaeological artifacts–each find was collected and its position recorded in 3D standard.

- Macrofossils–each bone, shell and charcoal detected at the site was collected and its position recorded in 3D standard.

- Paleontological samples–approximately 5-liter samples of the fine fraction of sediments (macrofossils collected separately, larger clasts removed by hands) were collected from each square, separately for each layer from each 5- or 10-cm interval. The samples were then wet-sieved through a 0.5-mm mesh, and any paleontological material was picked by hand.

- Geological samples– 36 samples of sediment with undisturbed grain size composition (i.e., fine fraction collected together with associating pebbles, cobbles and boulders), approximately 7 liters each, were taken from the walls of archaeological trenches according to methodology by Madeyska-Niklewska[61]. At least one sample was taken from each layer if possible, and in the case of layers with greater thickness, several samples were taken from one layer. The details on sample locations are provided in S1 File.

- Luminescence dating samples–four 0.5-liter samples of fine fraction were taken under dark conditions from selected layers with low amounts of limestone clasts and secured in opaque containers.

The collections of archaeological material and paleontological remains are stored at the Institute of Archaeology, Nicolaus Copernicus University in Toruń (Poland). Any paleontological finds from sieved samples (i.e., most of mollusk, rodent and bat remains) are stored under the inventory numbers of the samples (S2, S3 and S4 Files). Archaeological finds and macrofossils are stored under their individual repository numbers (S5 File).

## Lithological and geochemical analyses

Geological samples were dried and separated into fractions with a set of sieves (2, 4, 10, 20, 40 and 80 mm meshes). Macroscopic archaeological or paleontological material was removed by hand. The coarse fraction (>2 mm), composed almost exclusively of limestone clasts, was soaked in sodium phosphate solution for 24 h to dissolve clay aggregates and then washed with a shower on a 2-mm sieve. After drying, the material was weighted to calculate the amount of each fraction. The fine fraction (<2 mm) was homogenized by mixing and divided into smaller samples by quartering. Approximately 5-g samples of fine fractions were analyzed on a Malvern Mastersizer 2000 laser diffractometer with a HydroG adapter (analysis subcontracted to the Department of Geoecology and Palaeogeography, Maria Curie-Skłodowska University in Lublin). The procedure followed the widely accepted methodology[62].

Morphological classification of limestone clasts was performed on the >20 mm fraction, according to published methodology[61,63]. Four morphological classes were distinguished: A–sharp-edged (or angular) clasts; Ba–slightly smoothed (subangular) clasts with blunt edges; Bb–smoothed clasts with rounded edges (no edges recognizable), but with still-identifiable flat facets; C–highly smoothed (rounded-like) clasts with no recognizable edges or flat facets.

The chemical composition of the sediments was determined using approximately 1-g samples of homogenized <2 mm fraction. The analytical method was inductively coupled plasma mass spectrometry (ICP-MS) performed in the Bureau Veritas Minerals Laboratory (Vancouver, Canada) according to the published procedure[64]. The content of organic matter was

determined using another approximately 1-g aliquot of <2 mm fraction via oxygenation in 30% $H_2O_2$ for 24 h on a shaking plate[65]. If the reaction was still ongoing (bubbling or dark color), an additional portion of $H_2O_2$ was added for the next 24 h. The content of organic matter was calculated as the difference between the weight of the dry sample and that of the dry residue after the reaction.

## Chronometric dating

**Radiocarbon dating.** AMS radiocarbon dating was conducted in the Poznań Radiocarbon Laboratory (Poland). The dating material included snail shells, human and animal bones, charcoals and one example of organic char residue preserved on the internal surface of the ceramic pot[58]. In the case of snails, each dating was performed on one identifiable shell of *Isognomostoma isognomostomos*. Shells of *Discus ruderatus* were also dated, but due to their small size, several shells from the same layer, square meter and depth were mixed together. A small aliquot of each shell sample was checked with X-ray diffraction (at the Faculty of Geology, University of Warsaw, Poland) prior to dating to exclude any material with negative effect of aragonite-to-calcite recrystallization. Only samples evidencing pure aragonite were dated. Chemical pretreatment followed Brock et al.[66]. In the case of bones, the dated fraction was collagen extracted according to Goslar et al.[67]. Small aliquots of extracted collagen were analyzed with a CHNS elemental analyzer, and atomic C:N ratios was calculated to check the collagen quality. The range of 2.9–3.6 was assumed acceptable according to the literature[68,69]. For two samples, the collagen yield was too low for both elemental analysis and radiocarbon dating, and these samples were dated without the C:N ratio being confirmed. In the case of charcoals, cellulose was extracted according to AAA method[66].

All dates were calibrated using the OxCal v. 4.3 software package[70] versus the IntCal'13 radiocarbon calibration curve[71]. Calibrated dates are presented as ky BP (i.e., thousand years before 1950), as the ranges within a 94.5% probability interval. The OxCal v. 4.3 sequence model was applied to estimate the chronology of layer boundaries. It followed the procedure used by Krajcarz et al.[72], based on the widely accepted methodology[73,74].

**Luminescence dating.** The luminescence dating of samples was subcontracted to the Department of Geoecology and Palaeogeography, Maria Curie-Skłodowska University (Lublin). Post-IR IRSL$_{290}$ dating was performed on the polymineral fine grains (4–11 μm). The extraction procedure followed the accepted methodology[75]. Equivalent doses were determined using a post-IR IRSL protocol[76].

## Paleozoological analyses

Mollusks, rodents and bats are the most numerous groups of fossil animals in ShSIII and therefore were chosen for the paleozoological analysis to provide supportive paleoecological and paleoclimatic data for stratigraphy. In addition to these groups, the fossil material also comprises the remains of amphibians, reptiles, birds, insectivores and large mammals including humans. For the purpose of paleoecological analysis, layer 3 has been divided into two parts (3 lower and 3 upper) due to its great thickness and abundant rodent and mollusk remains.

**Identification of mollusk remains.** Standard methods[77–80] were applied for mollusk analysis. The shells collected during sieving were carefully cleaned. All completely preserved shells and their identifiable fragments were identified under a binocular microscope using taxonomical keys[81–83] and counted applying the schemes for broken individuals[77,80]. The number of individuals was counted separately for each layer. For some damaged individuals,

only the genus or family levels were determined, and the calcareous plates of slugs were counted together under the heading of Limacidae.

**Identification of rodent remains.** The rodent fossil assemblage consists of single bones and isolated teeth. Specimens were identified in terms of skeletal elements and species when possible. The taxonomical attribution of remains was based on the following diagnostic elements: first lower molars (Arvicolinae); mandibles, maxillae and isolated teeth (Muridae, Gliridae and Dipodidae); mandibles, maxillae, isolated teeth and diagnostic postcranial bones (*Cricetus cricetus*, *Sciurus vulgaris*). Remains were identified under a binocular microscope with use of the comparative collection of recent rodents and following the general criteria given in the identification keys[84,85]. The separation of *Microtus arvalis* and *M. agrestis* was based on the method given by Nadachowski[86,87]. The taxonomic classification followed Wilson and Reeder[88] for all taxa except for *Clethrionomys glareolus* (former *Myodes glareolus*) and *Lasiopodomys gregalis* (former *Microtus gregalis*), which were classified after later publications[89,90].

For each species, the number of identified specimens (NISP) and minimum number of individuals (MNI) were counted according to accepted method[91,92]. The MNI was counted separately for each layer.

**Identification of bat remains.** Specimens were identified, when possible up to the species level, on the basis of diagnostic traits of the hemimandible and/or the skull. In particular, the attribution of the taxonomic group was based on several different characteristics: the number of teeth, the shape of the hemimandible, the traits of the fourth lower and upper premolars and those of the upper and lower molars. Remains were identified and measured through a binocular microscope, comparing them with specimens belonging to a collection of recent bats. The general criteria given in the identification keys[93–95] were fulfilled, and the remains were classified taxonomically according to Wilson and Reeder[88]. The taphonomic parameters NISP and MNI were calculated according to the same method as for rodents.

**Paleoecological reconstruction.** Paleoecological inference was based on the environmental preferences of all mollusk, rodent and bat species found in the sediments. The structure of taphocoenoses and succession were presented on frequency bar diagrams. The diagrams were composed of both the absolute numbers of shells (or the MNI in the case of rodents) for samples containing less than 50 individuals and the percentages of the total sum for those with more than 50 individuals. In the case of bats, only the MNI values were shown due to low number of remains. Mollusk, rodent and bat assemblages were distinguished on the basis of their structure and composition, supported by statistical clustering analyses. The clustering was done for samples (layers) with more than 10 shells or with MNI greater than 10 by applying a paired group algorithm and Horn's overlap index for abundance data using the PAST software package, version 3.22[96,97]. Both the character and succession of the assemblages were used in the paleoecological and paleoclimatic reconstructions.

To reconstruct the changes in ecological diversity in the stratigraphic sequence, the assemblages were analyzed via application of the richness and the Simpson index of evenness. In the case of rodents, specimens identified as *M. agrestis/arvalis* were excluded, while *Apodemus sylvaticus/flavicollis* and *Apodemus* sp. were counted as one taxon (genus *Apodemus*). The number of taxa per sample (layer) is a measure of richness. The greater the number of taxa present in a sample, the 'richer' the sample. The Simpson 1-D index was calculated as $1 - \Sigma(n_i/n)^2$, where $n_i$ is the MNI of the particular taxon in an 'i-sample' (a layer) and n is the total number of MNIs in a sample (a layer)[97]. The index was calculated using the PAST software package, version 3.22. The index represents the probability that two individuals randomly selected from a sample will belong to different taxa.

To present the changes in the environment or landscape around the site, we used the environmental preferences of the species. Each mollusk taxon was assigned to one of four ecological groups[80]: F–shade-loving species; O–open-country species; M–mesophilous species; H–hygrophilous species. In the case of rodents, which usually inhabit a range of environments, we used the method of habitat weightings[98]. Each taxon was assigned to the habitat(s) where it can be found at present, with percentage weight(s) attributed for each of its habitat(s) (details are provided in S3 File). The rodent fauna of ShSIII was assigned to eight habitat types based on a habitat identification mode provided in the literature[99–101], with modifications for habitats typical for Central and Northern Europe. The used habitats were as follows: Tu–tundra; OA–open anthropogenic cultivated areas; OD–open dry (steppes); OH–open humid (meadows); OW–open woodland (shrubs, forest margins, steppe-forests); FT–temperate forest (broadleaf forests, mixed forests); FB–boreal forests (taiga); Wa–water bodies. Data on the species distribution and their preferred habitats were taken from Mitchell-Jones et al.[102] and the International Union for Conservation of Nature (IUCN) Red List maps[103,104,105–112,113–121].

Bats with their flight ability are more mobile animals and this restricts their usefulness as the indicators of local paleoenvironment. However, some species which are sedentary and use old trees as summer roosts (i.e., *Barbastella barbastellus*, *Myotis bechsteinii* and *Plecotus auritus*) clearly indicate the presence of old-growth forests in the vicinity[122–124]. Bats hunting near water (*M. dasycneme* and *M. daubentonii*) indicate the presence of open water bodies [125,126].

## Results

### Lithostratigraphic sequence

The lithostratigraphic sequence comprises fifteen layers, some of them being lateral variants. A characteristic feature of ShSIII is the lithological dissimilarity of sediments between different parts of the site, despite its small size (Figs 4 and 5). Three lithostratigraphic zones could be distinguished on the basis of these dissimilarities: entrance, central and rear. They were recognized at the squares of archaeological grid as follows: entrance zone–C/4, D/4, E/4, and partially C/5; central zone–the remaining area of C/5 and entire D/5, E/5, D/6, E/6, F/6 and F/7; rear zone–C/8 and C/9 (see also Fig 3). Detail lithological and geochemical parameters are provided in S1 File.

**Entrance zone.** The lower parts of the sedimentary sequence are similar at the entrance and central zones (Fig 4). The sediments of the upper part in the entrance zone are darker and more enriched in humus, as a result of the greater influence of external conditions and pedogenic processes. The total thickness of the sequence reaches 260 cm.

**Layer 9** is a limestone debris composed of coarse sharp-edged clasts. The matrix is a yellowish brown to brown (Munsell color: dry 10YR 5/4-6/4, moist 10YR 5/3-5/4) silty loam to sandy silt. The limestone bedrock below the layer is cracked into angular blocks, and this regolith passes into layer 9 gradually, which is marked by decreasing size of blocks and occurrence of matrix. The Ca content is very high in the fine fraction, particularly in the lower part of the layer, where it reaches greater than 30% (Fig 5). This strongly suggests that the fine fraction is mostly composed of physically disintegrated limestone.

**Layer 7** is a pale brown to brown (dry 10YR 6/3, moist 10YR 4/3) sandy silt. It is a massive (structureless) loess-like sediment with angular limestone clasts. Layer 9 passes indistinctly into layer 7, and the transition between the layers is marked by a gradual color change and decreasing amount of limestone clasts. The content of Ca in the fine fraction is very low (Fig 5), which indicates the allogenic source of material. At squares C/5 and partially D/5, a lens of

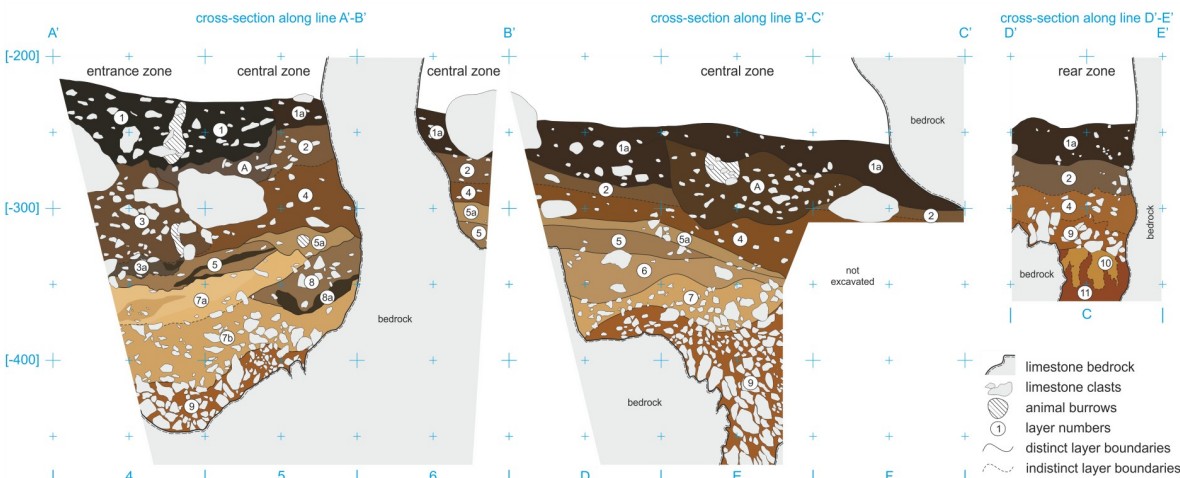

**Fig 4. The representative geological cross-sections through the sediments of ShSIII.** Natural colors of moist sediments are shown (Munsell colors are provided in text). Vertical scale is in centimeters below the site datum, which is at the elevation of around 447 m a.s.l. Location of cross-sections is shown in Fig 3.

dark silt (called layer 8) occurs in the upper part of layer 7. This intercalation enables dividing the layer locally into sublayers 7a (above layer 8) and 7b (below layer 8).

**Layer 8** is a grayish brown to dark grayish brown (dry 2.5Y 5/2, moist 10YR 4/2) sandy silt. The texture is similar to that of layer 7, except for the dark color and the presence of charcoals inside layer 8.

**Layer 3a** is a grayish brown to black (dry 10YR 5/2, moist 10YR 2/1) silt. The amount of limestone clasts is greater than in layer 7. Angular blocks predominate, but smoothed clasts

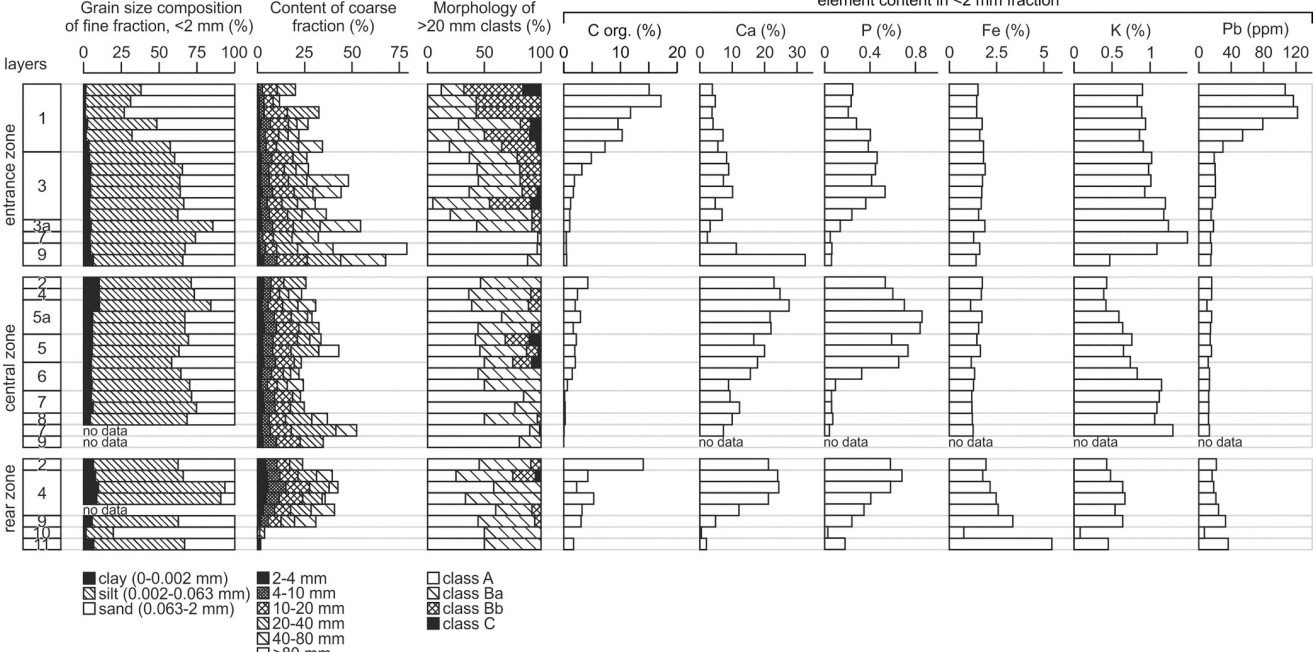

**Fig 5. Basic lithological and geochemical parameters of the sediments of ShSIII.** The data have been arranged by zones, layers and order of samples. For details see S1 File.

also occur. The concentration of P increases upward starting from layer 3a (Fig 5), indicating the increasing impact of zoogenic activity on the accumulation. The lower boundary is blurred, and the layer passes downward gradually into layer 7. Close to the walls of the rockshelter, the layer exhibits greater thickness, and its bottom is situated at lower elevation.

**Layer 3** is a dark brown to very dark grayish brown (dry 10YR 3/2-4/3, moist 10YR 2/2-3/3) sandy silt. The amount and size of limestone clasts is lower here, and smoothed clasts occur in greater number (Fig 5). However, outside of the dripline (square meters C/4 and partially D/4), the large limestone blocks occur in the layer. Their surfaces and edges are smoothed. Indistinct trough cross-stratification is marked in the bottom part at square E/4.

**Layer 1** is a very dark gray to black (dry 10YR 3/1, moist 2.5Y 2/1) silty sand. Limestone debris becomes less abundant and finer upward, and angular clasts are almost lacking, while smoothed, rounded-like clasts are relatively abundant. This layer exhibits unusually high concentrations of Pb and other toxic metals (Zn, Cd, Ag, Sb, Bi, Sn, see Fig 5), suggesting an increased impact of human activity. Another characteristic of the layer is a high sand fraction.

**Central zone.** The lithological variability is the greatest in this area, and the stratigraphy is the most complex, with up to eight distinct strata preserved in superposition (Fig 4). The lower part of the sequence shares lithological characteristics with the entrance zone, but in the upper part, the sediments are different. The sedimentary fill achieves a thickness similar to that of the sediments in the entrance zone (230 cm at maximum).

The sequence starts with a limestone debris containing very pale brown to yellowish brown (dry 10YR 8/3, moist 10YR 5/6) silty to sandy matrix (called **layer 9**), which is a continuation of layer 9 from the entrance zone. The clasts are finer here, but angular shapes still predominate (Fig 5). The layer reaches the greatest thickness (110 cm) at squares E/6, F/6 and F/7, where it fills the narrow and deep depression of the bedrock, probably a remnant of a vadose canyon.

The overlying sediment (called **layer 7**) is a continuation of layer 7 from the entrance zone. It is a yellowish brown to pale brown and brown (dry 10YR 6/3, moist 10YR 4/3–5/4) loess-like sandy silt with limestone clasts. The clasts are mostly angular and become less abundant and finer upward (Fig 5). The concentration of Ca in the fine fraction is very low (Fig 5), which suggests an allochthonous source of fine material. The lower boundary of this layer is indistinct, and layer 9 passes gradually into layer 7. Similar to the entrance zone, locally (at square D/5, where layer 8 occurs), the layer can be divided into sublayers 7a (above layer 8) and 7b (below). Sublayer 7a has a lighter, more yellowish color (dry 10YR 6/3, moist 10YR 5/4) and lower amount of limestone clasts compared with sublayer 7b (color: dry 10YR 6/3, moist 10YR 4/3). The lower boundary of sublayer 7a is not always readable, but locally, especially in contact with layer 8, it is sharp and erosional.

**Layer 8** is probably a continuation of layer 8 from the entrance zone. It is a grayish brown to dark grayish brown (dry 2.5Y 5/2, moist 10YR 4/2) sandy silt. It occurs in restricted area, limited to square D/5. The layer seems to fill the shallow depressions in sublayer 7b. The remains of a hearth are preserved in the lower part of layer 8 and labeled as sublayer 8a.

**Layer 6** is a light brownish gray to dark grayish brown (dry 10YR 6/2-6/3, moist 10YR 4/2–2.5Y 5/2) sandy silt. The amount, size and morphology of limestone clasts are similar to layer 7 (Fig 5). The concentrations of Ca and P increase from layer 6 upward, recording the increasing importance of autogenic deposition and zoogenic activity. The lower boundary of layer 6 is sharp and undulating. Internal sedimentary structures in a form of trough cross-stratification are weakly marked inside the layer.

**Layer 5** is a grayish brown to dark grayish brown (dry 10YR 5/2-5/3, moist 10YR 4/2-4/3) silty loam to sandy silt. The amount of limestone debris is greater than in layer 6 (Fig 5). The clasts are smoother, and even rounded-like ones occur. The lower boundary is discordant,

sharp and undulating. The trough cross-stratification is locally distinct, with bedding inclined toward the entrance.

**Layer 5a** is a whitish to grayish brown and dark grayish brown (dry 10YR 5/2-8/2, moist 10YR 4/3-6/3) massive silty loam to clay silt. The sediment is clearly lighter in color than those of the lower strata. Its lower boundary is sharp, and the layer lies discordantly on lower sediments. Locally, layers 5 and 6 are not preserved, and layer 5a lies directly on layer 7. The amount of limestone clasts is relatively low, and the clasts are fine and smoothed (Fig 5). The concentrations of Ca and P are among the highest in the entire sequence. The layer is not continuous and is lacking at square meters D/5, D/6 and E/6.

**Layer 4** is a very pale brown to brown (dry 10YR 7/3, moist 10YR 5/3) massive silty loam. The amount of limestone clasts is low; their morphology is similar to that in layer 5a (Fig 5). The clay content increases in the upper part of layer 5a and stays high in layer 4 (Fig 5). The concentration of Ca is similar to that in layer 5a, whereas the concentration of P is slightly lower (Fig 5). The lower boundary is indistinct and gradual.

**Layer 2** is similar to layer 4 in terms of the grain size composition and limestone clast morphology (Fig 5) but differs in color, being more grayish (dry 2.5Y 7/2, moist 2.5Y 4/2). The concentrations of Ca and P are still relatively high but lower than in layer 4. The boundary between layers 2 and 4 is indistinct and gradual.

The sequence is topped with **layer 1a**, which is a lateral variant of layer 1 from the entrance zone. This sediment is more brownish than layer 1 (dry 2.5Y 3/2, moist 2.5Y 2/2). The thickness of layer 1-1a decreases toward the rockshelter interior, such that the maximum thickness of layer 1 is 60 cm and that of layer 1a is up to approximately 30 cm. Upper part of layer 1a covers the archaeological feature (layer A), and due to lithological similarities of these two units, the boundaries between them are difficult to follow.

**Rear zone.** The sequence in the rear zone has a reduced thickness, achieving 120 cm at maximum (Fig 4). Sediments of this zone were recognized in a limited area of approximately 1 m$^2$. The rockshelter is shaded, and sunlight does not reach this area. The corridor declines downward to the south and becomes narrower, down to the width not accessible for penetration by an adult human. The concentration of K is low in this zone, which indicates lower infiltration from the top strata, possibly as a result of low rainwater supply in the deeper part of the rockshelter.

The sequence starts with **layer 11**, which is a complex stratum of bright reddish-brown color. The average grain size composition indicates a silty loam texture, but the layer is composed of numerous diapirs of brown (dry 7.5YR 5/4, moist 7.5YR 5/4) clay and lenses of reddish brown to strong brown (dry 7.5YR 6/6, moist 7.5YR 5/6) silty sand. This sandy component is most likely an admixture from layer 10. Limestone clasts are absent. The sediment is strongly enriched in Fe and depleted in Ca (Fig 5), what indicates an allogenic origin. High concentrations of Fe and other geochemically related metals (Mn, Co, Zn, Pb, Cu, Mo, Th, Cd, Sb, and REE) are characteristic features of this sediment and are responsible for its reddish color. The thickness is greater than 30 cm.

**Layer 10** is a yellow to brownish yellow (dry 10YR 7/5, moist 10YR 6/7) silty sand. Sandy grains are composed mostly of quartz. Such a mineral composition, together with the lack of limestone clasts and very low concentration of Ca (Fig 5), indicates an allogenic origin. The sediment is preserved only in pocket-like or tongue-like structures immersed in layer 11. The contact between the two layers, although not in a primary position, is sharp, what indicates the erosional boundary at the bottom of layer 10. The original thickness of the layer is difficult to establish due to disturbances, but it can be estimated to be greater than 5 cm.

**Layer 9** is a 15-cm-thick horizon of coarse subangular limestone blocks, with yellowish brown to dark yellowish brown (dry 10YR 5/3, moist 10YR 4/4) silty loam filling the space

between blocks. The lower boundary is sharp and erosional. The vertical orientation typical for layers 11 and 10 is not visible here. High concentrations of trace metals, similar to those exhibited by layer 11 (Fig 5), indicate that layer 11 served as important source of material. This layer most likely forms the lateral continuation of layer 9 from the entrance and central zones.

That layer passes upward gradually into **layer 4**. This is a very pale brown to yellowish brown (dry 10YR 7/3-7/4, moist 10YR 5/4-5/5) silt or clay silt, passing upward into silty loam. This layer is a lateral continuation of layer 4 from the central zone. The concentrations of Ca and P are high and increase toward the top of the layer.

**Layers 2 and 1a** are lateral continuations of layers 2 and 1a, respectively, from the entrance zone and share lithological characteristics with those strata.

**Anthropogenic layers.** Two anthropogenic strata were recognized between the natural strata and were labeled as sublayer 8a of layer 8 and layer A.

**Sublayer 8a** is a black (dry 10YR 4/1, moist 10YR 2/1) sandy silt. It occurs in the lower part of layer 8 and represents its darker variant. The sublayer has a form of a thin lens with a thickness of approximately 10 cm and lies at the bottom of a shallow depression dug into layer 8 and reaching sublayer 7b. The presence of charcoals indicates that this deposit is a remnant of a hearth.

**Layer A** is a sediment of intermediate characteristics between layers 1, 1a, 2 and 3. It is a dark brown to very dark grayish brown (dry 10YR 3/1-4/3, moist 10YR 2/2-3/2) silty sand. It forms a backfill of an anthropogenic feature. Its lower boundary is sharp and distinct. Stratigraphically, the feature is situated inside layer 1-1a, as its walls cut layer 1 or 1a, and at some square meters, it is covered by the upper part of layer 1a, with an indistinct boundary between the layers. The bottom of the feature reaches layer 2 or 3 (at different square meters) and occasionally layers 4 and 5a.

## Chronometric data

A total of 29 radiocarbon dates are accessible for the ShSIII (Table 1). The dates cover the interval from 14.1 to 0.6 ky BP, i.e., almost the entire Holocene and Late Glacial of the Last Glaciation (Fig 6). In case of bones, the atomic C:N ratio in extracted collagen is in the widely accepted 2.9–3.6 range[68,69] for all tested samples (Table 1; only two samples yielded too low amounts of collagen to check the C:N ratio), confirming that the analytical fraction was pure. Four TL dates were also obtained for layers 7b, 6 and 5a (Table 2).

To build the sequence model in OxCal, we used only the radiocarbon dates from the central zone, to achieve the data for a single stratigraphic column and avoid problems due to lateral correlations. Moreover, the central zone delivered most of the dates, while only two radiocarbon dates were obtained from material from the entrance zone. The archaeological feature (layer A) was excluded because its backfill includes redeposited material from the lower layers and may disturb the model of deposition. A date from layer 1a (Poz-53301) was also excluded, as this date was obtained from a human bone, which was most likely redeposited from the grave, i.e. from layer A. The model including all dates from the central zone did not run until the chronologically discontinuous dates were removed. The dates in disorder were mostly those from layers 4, 5a and 5. There are several possibilities for selecting dates for removal. The least-invasive method is to remove three dates, Poz-61237, Poz-61305 and Poz-53303 (marked in Fig 6 as "intrusive material?"), and this method was applied to build the model (Table 3).

The dates from layer A (backfill of archaeological feature) and from a human bone from layer 1a (Poz-53301) were used to build a separate sequence model to determine the boundaries of the phases of human activity. The dates were attributed in an arbitral manner to three phases of human activity (PHA): PHA 2 –Bronze Age; PHA 3 –pre-Roman to Roman Period;

**Table 1. Radiocarbon dates for material form ShSIII.** New data and all published dates[58–60,127] are provided.

| Layer | Lab. No. | C-14 date | Material | Inv. No. | Square | Zone | Analytical fraction (atomic C:N ratio for collagen) |
|---|---|---|---|---|---|---|---|
| 1a | Poz-53301 | 645±30 | bone, *Homo sapiens* | W-69 | E6 | central | collagen (3.00) |
| A | Poz-56204 | 730±30 | bone, *Homo sapiens* | W-1 | D5 | central | collagen (3.23) |
| A | Poz-56208 | 2080±30[a] | organic residue in the ceramic vessel | W-382 | F7 | central | bulk |
| A | Poz-53302 | 2190±30[a] | bone, *Ursus arctos* | W-123 | E5 | central | collagen (2.99) |
| A | Poz-53300 | 2270±30[b,d] | bone, *Felis silvestris* | W-93 | E5 | central | collagen (2.97) |
| 1a | Poz-114532 | 2430±30 | bone, *Felis silvestris* | W-63 | D5 | central | collagen (3.48) |
| 2 | Poz-51326 | 2810±30[c,d] | bone, *Felis silvestris* | W-183 | E6 | central | collagen (3.25) |
| A | Poz-56206 | 3665±35 | bone, *Canis familiaris* | W-239 | E6 | central | collagen (n.d.) |
| A | Poz-53308 | 3745±30 | bone, *Canis familiaris* | W-92 | E5-6 | central | collagen (3.10) |
| A | Poz-56205 | 3770±35 | bone, *Homo sapiens* | M-98 | E6 | central | collagen (2.90) |
| 4 | Poz-53303 | 4360±35 | bone, *Meles meles* | W-280 | E6 | central | collagen (3.12) |
| 2 | Poz-51327 | 4475±35[c,d] | bone, *Felis lybica/catus* | W-195 | E6 | central | collagen (3.26) |
| 4 | Poz-53305 | 4895±35 | charcoal | P-40 | E6 | central | cellulose |
| 4 | Poz-61304 | 4940±35 | shell, *Isognomostoma isognomostomos* | - | D-E5-6 | central | aragonite |
| 5a | Poz-61305 | 5870±40 | shell, *Isognomostoma isognomostomos* | - | E5 | central | aragonite |
| 4 | Poz-114530 | 6160±40 | bone, *Felis lybica/catus* | M-6 | E6 | central | collagen (3.10) |
| 4 | Poz-51325 | 7020±40[d] | bone, *Felis silvestris* | M-8 | D6 | central | collagen (3.29) |
| 4 | Poz-51328 | 7400±40 | bone, *Martes martes* | W-271 | E6 | central | collagen (3.07) |
| 3a | Poz-61306 | 7500±40 | shell, *Isognomostoma isognomostomos* | - | E4 | entrance | aragonite |
| 5 | Poz-61237 | 7570±40 | 8 shells, *Discus ruderatus* | - | E4-5 | central | aragonite |
| 4 | Poz-61235 | 8270±40 | 3 shells, *Discus ruderatus* | - | D-E5-6 | central | aragonite |
| 5a | Poz-61236 | 8450±40 | 11 shells, *Discus ruderatus* | - | E5 | central | aragonite |
| 6 | Poz-61240 | 8990±40 | 8 shells, *Discus ruderatus* | - | E5 | central | aragonite |
| 5 | Poz-61238 | 9090±50 | 9 shells, *Discus ruderatus* | - | D-E4-5 | central | aragonite |
| 8a | Poz-53307 | 10,790±70[c] | charcoal | P-62 | D6 | central | cellulose |
| 8a | Poz-53306 | 11,000±60 | charcoal | P-53 | D5 | central | cellulose |
| 7b | Poz-53304 | 11,730±60 | bone, *Lepus* sp. | W-290 | D4 | entrance | collagen (n.d.) |
| 8 | Poz-61241 | 11,870±60 | 3 shells, *Discus ruderatus* | - | D4 | central | aragonite |
| 7b | Poz-56207 | 12,110±60[c] | bone, *Alces alces* | W-455 | C5 | central | collagen (3.09) |

95.4% probability ranges are shown for calibration.

[a] after Sudoł et al.[58]

[b] after Krajcarz et al.[60]

[c] after Sudoł et al.[59]

[d] after Baca et al.[127]

n.d.–not determined due to low collagen yield

and PHA 4 –Late Middle Ages (PHA 1 was reserved for the Late Paleolithic phase). The model for PHA boundaries gave very good overall agreement index $A_{overall}$ = 96.4%. The model for the boundaries of the natural layers gave weaker agreement index, $A_{overall}$ = 78.8%, yet still in the acceptable range which is >60%[73,74]. This weaker agreement is a consequence of partial overlapping between the extreme dates from layers 2 and 4; 4 and 5a; and especially 5 and 6. The outlier with the lowest agreement index is a date for layer 5 (Poz-61238, agreement index A = 39.7%). The obtained modeled boundaries are provided in Table 3.

## Malacological succession

The molluskan fauna from ShSIII is composed of 61 taxa of land snails, represented altogether by 10,295 individuals. Fifty-two species are accompanied by 7 taxa identified to the genus

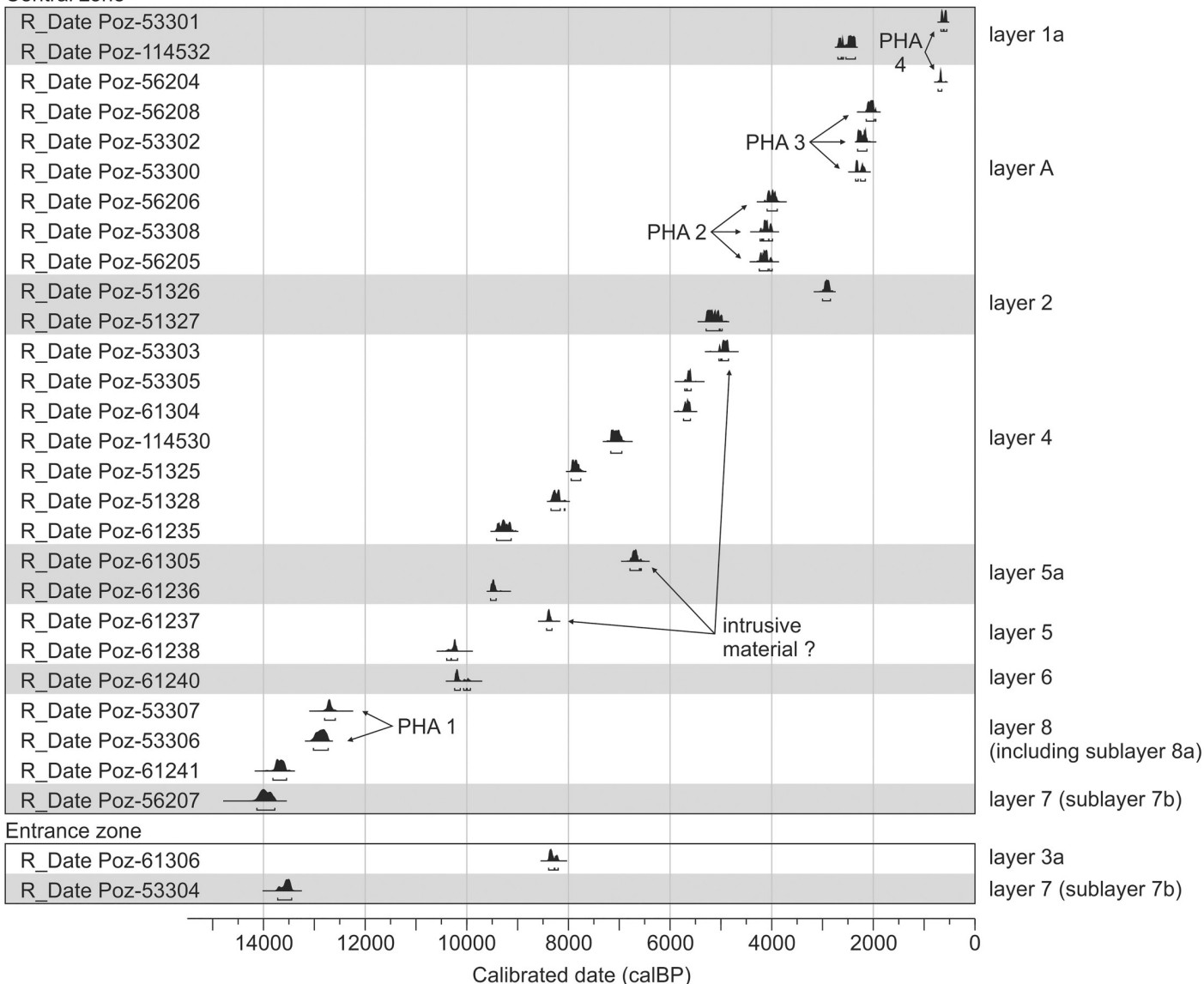

**Fig 6. Calibrated radiocarbon dates for layers in ShSIII.** Dates are arranged here by the order of layers. For details see Table 1. PHA are phases of human activity, described in the section "Chronometric data".

**Table 2. Optically stimulated infra-red luminescence (IRSL) dates of sediments from ShSIII.**

| Layer | Lab. No. | Annual doze (Gy/ka) | Equivalent doze (Gy) | Age (ky BP) | Inv. No. | Square | Zone |
|---|---|---|---|---|---|---|---|
| 5a | Lub-5523 | 1.37±0.07 | 44.3±2.4 | 32.3±2.4 | P-TL1 | D6 | central |
| 6 | Lub-5524 | 2.27±0.11 | 43.0±1.7 | 18.9±1.2 | P-TL2 | D6 | central |
| 7b | Lub-5526 | 2.22±0.11 | 47.9±1.5 | 21.6±1.3 | P-TL4 | D5 | central |
| 7b | Lub-5525 | 2.69±0.13 | 65.7±2.1 | 28.9±1.7 | P-TL3 | D4 | entrance |

**Table 3. The age of the boundaries between phases for ShSIII modelled in OxCal.**

| Boundaries | modelled boundaries (years BP) | | C (%) |
|---|---|---|---|
| | for 68.2% | for 95.4% | |
| phases of human activity | | | |
| upper boundary of PHA 4 | 670–420 | 680–-260 | 97.3 |
| between PHA 3 and PHA 4 | 2020–650 | 2050–650 | 98.5 |
| between PHA 2 and PHA 3 | 4100–3670 | 4140–2250 | 97.6 |
| lower boundary of PHA 2 | 4310–4020 | 4800–3990 | 95.6 |
| natural layers | | | |
| between layers 1a and 2 | 2870–2470 | 2920–2400 | 99.1 |
| between layers 2 and 4 | 5610–5290 | 5640–5100 | 99.6 |
| between layers 4 and 5a | 9480–9300 | 9510–9180 | 99.6 |
| between layers 5a and 5 | 10220–9470 | 10220–9470 | 98.4 |
| between layers 5 and 6 | 10240–10180 | 10250–9980 | 99.8 |
| between layers 6 and 8[a] | 12730–10180 | 12740–10180 | 95.9 |
| between layers 8 and 7b | 13950–13700 | 14060–13610 | 99.5 |
| lower boundary of layer 7b | 14360–13820 | 16210–13760 | 95.1 |

Ages are shown separately for 68.2% and 95.4% probabilities. In case of the natural layers (layers 7b, 8, 6, 5, 5a, 4, 2 and 1a), the modelled boundaries represent the boundaries between the natural strata. In case of the phases of human activity (layers 1a and A), the modelled boundaries represent boundaries between the phases of human activity (PHA). *C* is an OxCal measure of convergence.

[a] no dates were achieved for sublayer 7a situated between layers 6 and 8, so the modelled chronology of sublayer 7a falls within this boundary

level, one to the family level (Clausiliidae), and slug plates represent a collective group of Limacidae (S2 File). Malacofauna represents four ecological groups (according to widely accepted subdivision[80]), with 30 shade-loving species (group F), 8 open-country species (group O), 13 mesophilous species (group M) and 1 hygrophilous species *Carychium minimum* (group H). There are only slight differences in malacological composition between the central and entrance zones (S2 File), but they differ in proportions between taxa and structure of the assemblages.

No gastropods were found in the lower part of sedimentary sequence (in layers 11, 10 and 9). In the above-lying layers, the mollusk community is bipartite. Only few taxa and up to 60 individuals altogether are noted in layers 7b, 8 and 7a, whereas the higher part of the sequence is characterized by rich malacocoenoses comprising up to 46 taxa and 2,181 individuals in a single layer.

Based on both the cluster analysis and the fauna composition, four mollusk assemblages were distinguished in ShSIII. The nominal species are not the most abundant but rather the most characteristic of the population, which enables wider comparisons on the regional scale [35,36,41,128–130].

**Assemblage Vt.** The sequence starts with the assemblage with *Vallonia tenuilabris* (Vt), including layers 7b, 8, and 7a in the central zone and layer 8 in the entrance zone; possibly also layers 7a and 7b in the entrance zone belong to the same assemblage. This poor malacocoenosis contains from 5 to 60 shells per layer (Fig 7), mostly of open-country gastropods, which constitute 35–73% of all individuals. *Vallonia tenuilabris*, a glacial relic that occupies the open areas of so-called "cold-steppes", the alpine grasslands, taiga, hemiboreal forest and wooded fens[77,131–134], is the most numerous in the central zone, being accompanied by the shade-

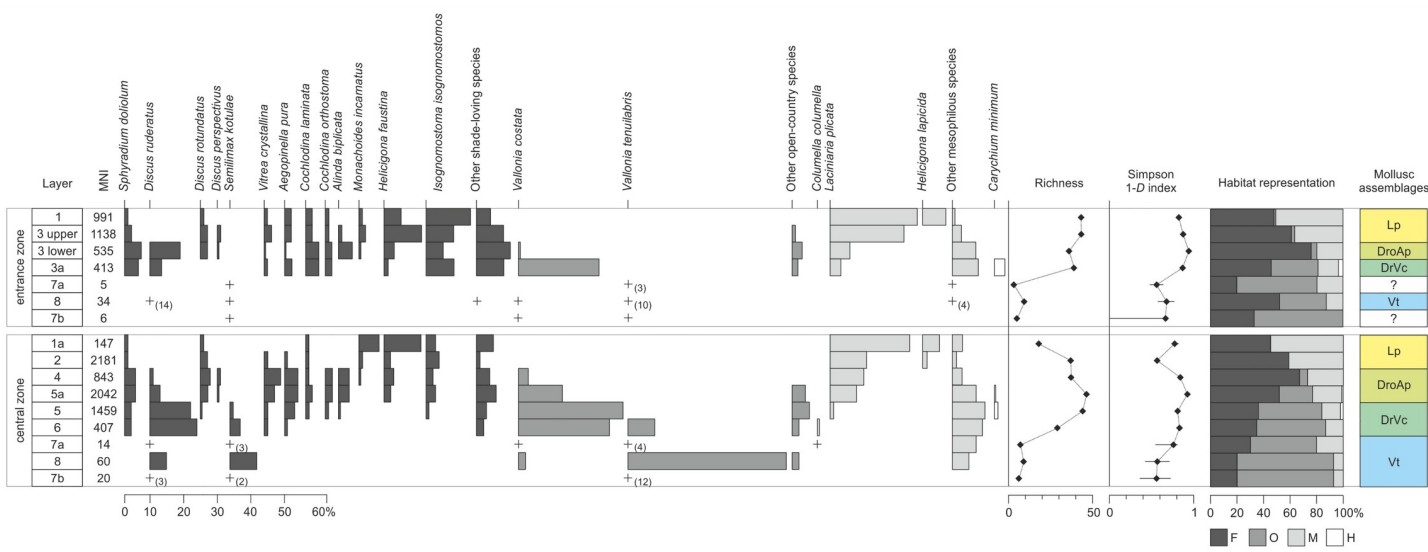

**Fig 7. Paleoecological indicators for ShSIII based on mollusk remains.** Data are arranged by the zones and the order of layers. Representation of taxa reflects the percent of individuals. Dark gray bars are used for shade-loving, medium dark for open-country, light gray for mesophilous and white for hygrophilous taxa. "+" means that the taxon is represented in the given layer but the total MNI for the layer is lower than 50, so the representation is not statistical. Numbers in parentheses indicate the number of individuals higher than 1 for statistically non-representative layers. Measures of the paleoecological diversity (richness and Simpson index), representation of habitats and attribution of layers to mollusk assemblages are also shown.

loving species *Discus ruderatus* and *Semilimax kotulae*, both characteristic of a cool and continental climate. In the entrance zone, *D. ruderatus* slightly outnumber *V. tenuilabris* (Fig 7).

**Assemblage DrVc.** In layers 6, 5 and 3a, another assemblage dominated by *Discus ruderatus* and *Vallonia costata* (DrVc) can be distinguished (Fig 7). *D. ruderatus* represents taiga-type coniferous forests, whereas *V. costata* is typical of dry and open habitats and predominates in the sequence. In the central zone (layers 6 and 5) glacial relics *V. tenuilabris* and *S. kotulae* are still important components of the assemblage, but they disappear in the entrance zone (layer 3a). Moreover, this assemblage records a first period of predominance of shade-demanding species in the sequence (Fig 7). Mesophilous snails are mostly represented by *Carychium tridentatum*, associated with permanent humid conditions [83] (S2 File).

**Assemblage DroAp.** A rich assemblage with *Discus rotundatus* and *Aegopinella pura* (DroAp) can be distinguished in layers 5a and 4 of the central zone and the lower part of layer 3 in the entrance zone. This assemblage is represented mostly by forest gastropods (52–75% of all individuals) typical of warm and humid conditions, namely, *D. rotundatus*, *A. pura*, *Discus perspectivus*, *Balea biplicata*, *Helicigona faustina* and *Isognomostoma isognomostomos*. *V. costata* gradually disappears, and the open-country snails become outnumbered by mesophilous species, mainly *Laciniaria plicata* (Fig 7). The occurrence of *V. elata*, unknown in the contemporary fauna of the region, is worth noting.

**Assemblage Lp.** The topmost part of the sequence (the upper part of layer 3 and layers 2, 1a and 1) is occupied by forest and mesophilous taxa, with the most numerous being *Laciniaria plicata* (Lp). Gastropods of open habitats disappear in the central zone and are of secondary importance in the entrance zone. The forest fauna still comprise a large fraction of the total fauna (45–61% of all individuals) but are gradually replaced by snails of wide ecological tolerance, which are the most abundant in layers 1a and 1 (up to 55% of all individuals) (Fig 7). *L. plicata*, which is typical of humid rocks in open environments, is accompanied by *Helicigona lapicida*, which often hides in deep crevices in shady rocky substrate; *H. faustina*, which

is common in forest slopes and humid shady rocks; and *I. isognomostomos*, which is characteristic of humid mountain forests (Fig 7 and S2 File).

## Rodent succession

The fossil assemblage includes 1,716 identified specimens, corresponding to a minimum of 901 individuals (sum of MNIs for particular layers) and representing at least 19 taxa, including 17 taxa identified to the species level (S3 File). There are no taxonomical differences in the rodent composition between the central and entrance zones. The sediments of the rear zone appeared as very poor in rodent remains and provided only 45 identified specimens in total. For this reason, the usefulness of fauna from the rear zone for statistical processing is not plausible.

No rodent remains were collected from the lowermost layers (11 and 10). Very poor collection (3 specimens) was found in layer 9. In the upper sequence, two main parts are distinct, which vary in terms of both taxonomic composition and number of specimens (S3 File). The lower part (layers 7 and 8) is characterized by a low abundance of rodent remains, low richness and poor diversity (Fig 8). Much richer taphocenosis is preserved in the upper part of the sequence. The number of individuals and number of taxa reach their maxima in layers 5, 5a, 4 and 3 (its lower part). The habitat representation recorded by rodent remains varies clearly between the lower and upper parts of the sequence. The representation of forest environments is low in layers 8 and 7, where the indications of an open tundra environment with the presence of water bodies predominate. The forest component increases in layer 6 (51%) and continues up to reach over 70% in the upper layers (with maximum in layers 4, 2 and the upper part of layer 3).

The cluster analysis of rodent assemblage from ShSIII reveals four distinct groups, which reflect the distinct faunal assemblages formed by the changes in the composition and structure of the species (Fig 8). These groups were named after the most abundant taxa.

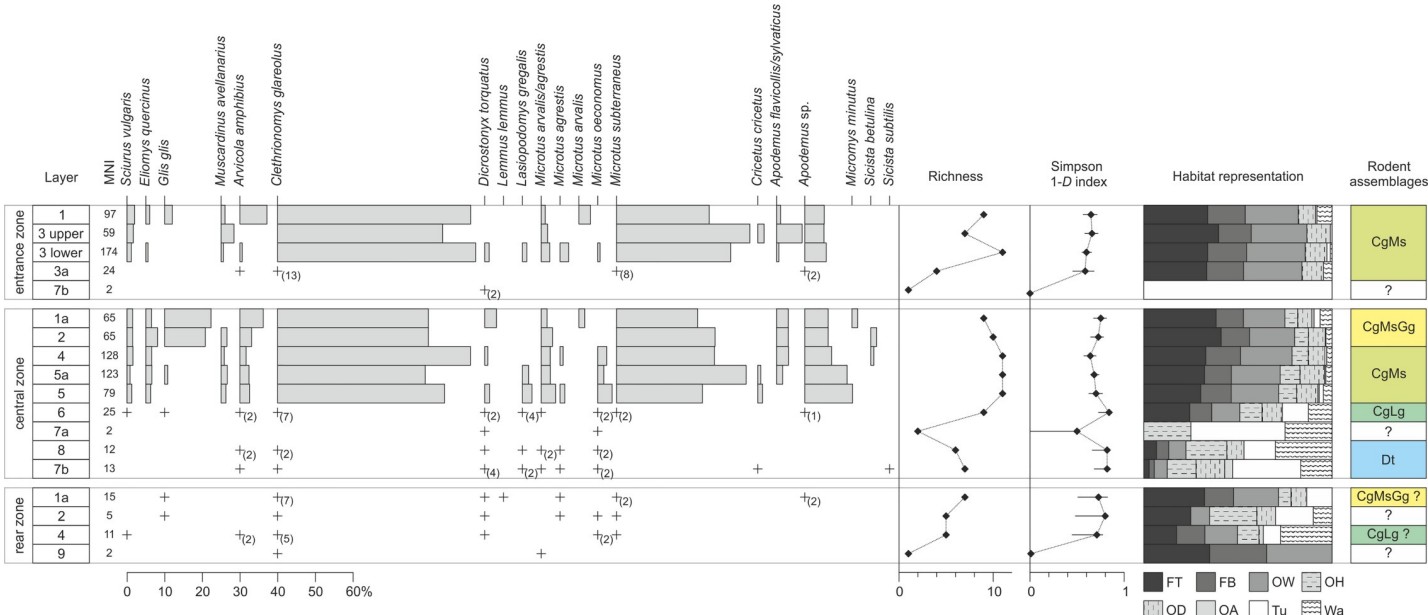

**Fig 8. Paleoecological indicators for ShSIII based on rodent remains.** The arrangement of figure follows Fig 7. Due to multiple habitats settled by individual taxa, the bars of taxa representation do not indicate the habitats and taxa are arranged systematically (by families and genera).

**Assemblage Dt.** This assemblage groups layer 8 and sublayer 7b from the central zone. Sublayer 7a from the central zone and layer 7b from the entrance zone possibly also belong to this assemblage. From a quantitative point of view, this is a very poor collection, giving altogether 38 remains (at least 29 individuals). The dominant taxon here is a collared lemming *Dicrostonyx torquatus* (30.8–100% of MNI for sublayers 7a and 7b), followed by the tundra vole *Microtus oeconomus* and European water vole *Arvicola amphibius* (Fig 8). Collared lemming is an inhabitant of cold climatic areas, while the two other voles prefer humid habitats with water bodies. The relatively high representation of the red vole *Clethrionomys glareolus* in layer 8, which is linked with woodlands and prefers densely vegetated clearings, suggests a possible temporal climatic improvement with development of woodland vegetation. Inhabitants of open dry environments (steppes), i.e., *Cricetus cricetus* and *Sicista subtilis*, were recorded in sublayer 7b.

**Assemblage CgLg.** Layer 6 forms a separate assemblage. The number of rodent remains is low (51 remains, at least 25 individuals). There is a well-distributed diversity, even if the number of species is low. The assemblage is dominated by *C. glareolus* (28% of MNI). The second-most-abundant species is *Lasiopodomys gregalis* (16%), a representative of open and cool habitats. Rodents adapted to cold or boggy environments (*D. torquatus*, *A. amphibius* and *M. oeconomus*) are still present. Thus, the layer records a sensible increase in the woodland component (*Apodemus* and *C. glareolus* constitute 30% of the MNI) (Fig 8). Layer 4 from the rear zone is grouped by cluster analysis with this assemblage, however, the number of specimens is low (15 remains, at least 11 individuals) and this clustering is disputable.

**Assemblage CgMs.** A rich assemblage dominated by *C. glareolus* and *M. subterraneus* is recorded in layers 3a, 3 and 1 in the entrance zone, layers 5, 5a and 4 in the central zone. Layer 2 from the rear zone is poor in rodent remains but possibly also represents this assemblage. The species of cold habitats disappear. There is a visible rise in the representation of taxa associated with woodland environments, with domination of the red vole, *C. glareolus*. This is the most abundant taxon in the whole upper part of the sequence. Starting from layer 5 in the central zone and 3 in the entrance zone upward, the red vole is followed by *M. subterraneus*, also an inhabitant of woodlands, preferring densely vegetated clearings. Mice of genus *Apodemus* also constitute a significant portion of the assemblage. The development of deciduous or mixed forest habitats is testified by the presence of three species of the Gliridae family, which continues in the upper layers of the sequence (Fig 8).

**Assemblage CgMsGg.** The layers of the uppermost part of the sequence in the central zone (2 and 1a) are characterized by another rodent assemblage. The species representation is similar to the lower assemblage CgMs, and the record of habitat conditions did not change significantly. The visible change is recorded by the higher participation of the edible dormouse *Glis glis*. In layer 1 of the entrance zone and layers 2 and 1a of the rear zone the presence of this species is also marked, which suggests their correlation with CgMsGg assemblage. However, the amount of edible dormouse remains is not as high there as in layers 2 and 1a in the central zone, and therefore the cluster analysis joins these layers with other assemblages (Fig 8 and S3 File).

## Chiropteran succession

The bat fossil assemblage includes 369 remains corresponding to a minimum of 246 individuals (sum of MNIs of different layers) and representing at least 24 taxa, including 13 species (for the specimens that were identified to the species level) and 6 groups (remains identified to the genus level or belonging to cryptic groups) (S4 File); the latter were not considered during the statistical analysis. For some specimens, identification of the species was not possible, since the

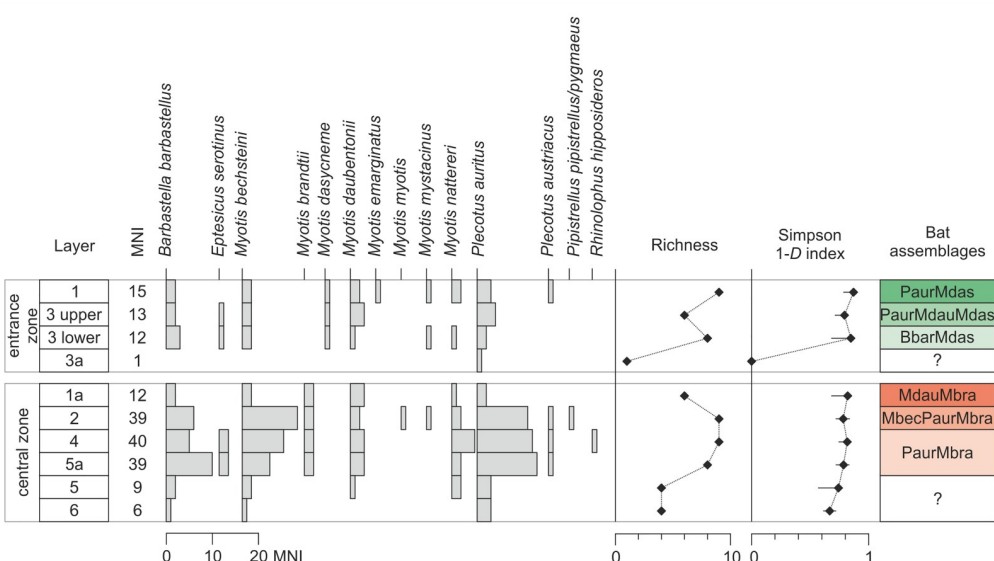

**Fig 9. Paleoecological indicators for ShSIII based on bat remains.** The arrangement of figure follows Fig 8.

morphological and dimensional characteristics required for a precise assessment were missing. The groups affected by this issues were *Plecotus auritus/austriacus*, *Pipistrellus pipistrellus/pygmaeus*, *Myotis mystacinus/brandtii/alcathoe/daubentonii*. Some remains (mainly fragmented bones) were identified just to the order level (e.g., Chiroptera).

No bat remains were found in the rear zone. There are differences in the taxonomical composition of bat assemblages between the central and entrance zones. The sediments of the entrance zone are poor in bat remains, with 41 bat specimens representing 10 taxa. The chiropteran remains are much more abundant in the sediments of the central zone, with 144 specimens belonging to 11 taxa. The presence of *Myotis dasycneme* characterizes the entrance zone exclusively, while *M. brandtii* occurs in the central zone instead. No bat remains were collected from the lowermost layers (11 to 7). Layers 5a, 4 and 2 yielded the highest number of bat remains (Fig 9). This increase in the presence of bats could be explained by the advance in the forest environment in the area. The cluster analysis of the bat assemblage revealed six distinct groups.

**Assemblage PaurMbra.** The group comprised by layers 4 and 5a contains the richest assemblage in terms of number of remains, even if the number of species does not differ from the overlying layer 2 (Fig 9). We found 79 individuals belonging to 9 species, for a total of 130 bone remains.

**Assemblage MbecPaurMbra.** Layer 2 in the central zone seems to be one of the richest layers in terms of a number of remains (NISP = 61, represented by 38 individuals belonging to 9 species). The only fragment attributable to *P. pipistrellus/pygmaeus* (the identification of the species is impossible due to a lack of the distinctive characters) was found in this layer.

**Assemblage MdauMbra.** This assemblage represents layer 1a from the central zone and is comparable to the one found in layer 1 from the entrance zone (i.e., the PaurMdas assemblage). A small number of remains (16 fragments) were found. They belong to 12 individuals of 6 different species (Fig 9).

**Assemblage BbarMdas.** The lower part of layer 3 from the entrance zone delivered few fragments (16) belonging to 12 individuals of 8 species. This assemblage does not differ considerably from the other layers of the entrance zone (Fig 9).

**Assemblage PaurMdauMdas.** This assemblage, belonging to the upper part of layer 3 in the entrance zone, includes a small number of specimens (16 bone fragments attributable to 13 individuals of 6 different species) (Fig 9).

**Assemblage PaurMdas.** This assemblage represents layer 1 from the entrance zone. From a quantitative point of view, this is a very poor collection, with only 16 remains belonging to at least 15 individuals (Fig 9). A predominant taxon cannot be defined.

## Discussion

### Deposition of sediments

The entire sedimentary sequence in ShSIII can be divided into six sedimentary series (Fig 10). The differences between series are marked in sedimentary structures (erosional surfaces and lamination), texture (mostly an abundance of limestone clasts) and geochemistry (mostly the amount of organic matter, Ca and P).

**Series I–reddish clays.** The oldest sediments are preserved in the rear zone as layer 11. This sediment shares similarities with reddish clays known from the lowermost parts of the sequences from caves such as Biśnik Cave (layers 20–23)[26], Nietoperzowa Cave (layer 17) [20], Tunel Wielki Cave (lower part of layer 1)[21], Cave in Dziadowa Skała (lower part of layer 1)[135] and Żabia Cave (lower part of sequence)[136]. These types of sediments were linked with Lower Pleistocene or Pliocene *terra rosa* soils, redeposited to the caves from the surface[137,138].

**Series II–sands.** Layer 10 has also been preserved only in the rear zone. The bright color of sand suggests a correlation with reddish or yellowish sands known from other caves of the region, where they also form a bottom part of the sequences. Such sandy deposits are known

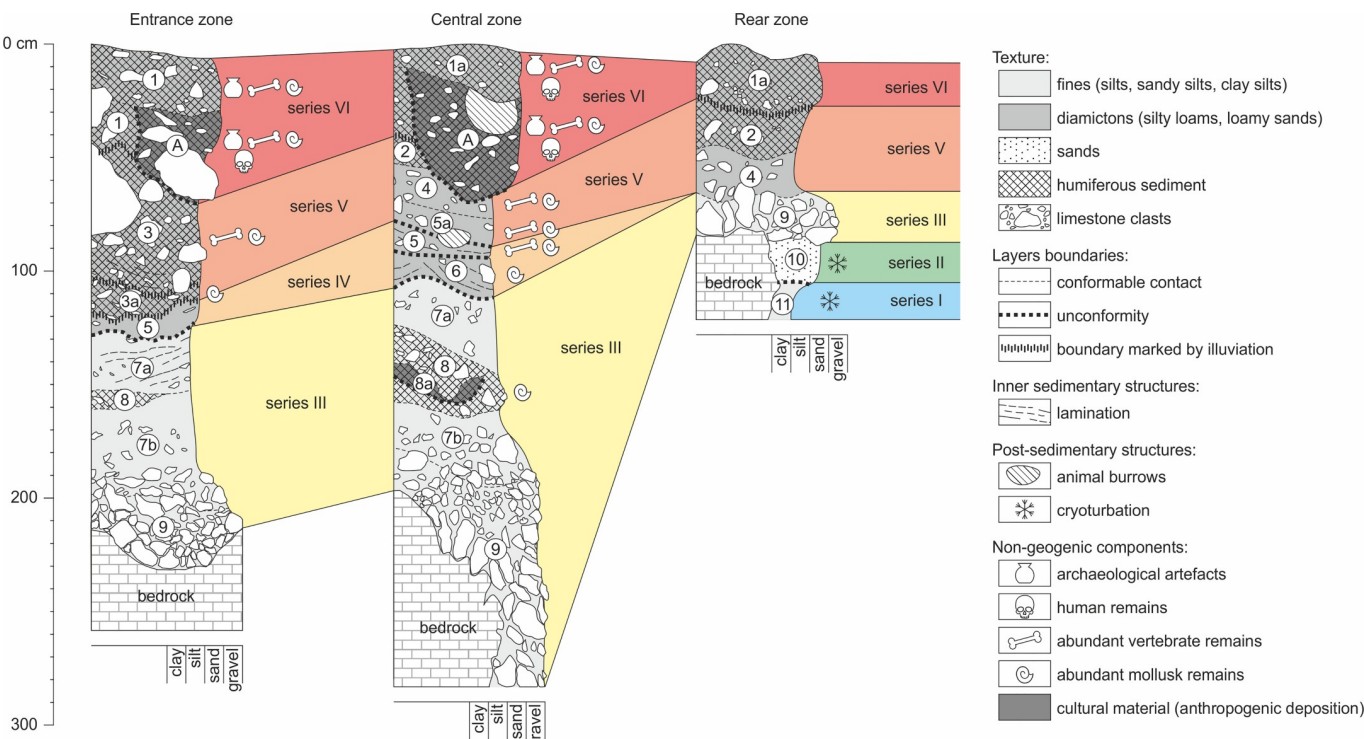

**Fig 10. Litho-stratigraphic logs of the Quaternary sediments of ShSIII.** Attribution of layers to sedimentary series and correlation between the zones of the site are shown.

from caves such as Nietoperzowa Cave (layer 17)[20], Koziarnia Cave (layer 21)[20], Tunel Wielki Cave (upper part of layer 1)[21], Perspektywiczna Cave (layer 11)[46], Cave in Dziadowa Skała (upper part of layer 1)[135] and Wylotne Rockshelter (layer 8)[20]. The accumulation of sands may be connected with Middle Pleistocene glaciations, when the area of Kraków-Częstochowa Upland entered the fluvioglacial regime[51,52,139], or with later colluvial redeposition of fluvioglacial sediment. Both layers 11 and 10 fill the narrow depression of the bedrock, a form which was called a "bottom rill"[140] and which is most likely the remnant of a vadose canyon. This depression was also noticed in the central zone, but here, the sediments resembling layers 10 or 11 have not been preserved. The depression is filled with limestone debris of layer 9, which also occurs in the rear zone, but with limited thickness and at higher elevation. This result indicates that erosional event happened between accumulation of layers 10 and 9, which was responsible for almost total removal of layers 10 and 11, except for the rear zone.

**Series III–limestone debris and loess.** Layer 9 records the cold climatic conditions. A high amount of angular limestone clasts with Ca-rich silty matrix indicates that physical weathering of the bedrock and consequential rockfall was the most important depositional factor, followed by physical weathering of the rockfall debris. An absolute lack of faunal remains additionally suggests very cold periglacial conditions. This sediment may be correlated with layer III from the nearby Shelter above the Zegar Cave[141], described as loess with limestone debris, with very sparse faunal remains. The reddish color of layer 9 suggests that layer 11 (and possibly also layer 10) served as an additional source of material for layer 9. Most probably, some remnants of layers 11 and 10 survived the erosion adhered to the walls or inside the niches and dropped down during the deposition of layer 9. A similar situation concerning the Lower Pleistocene red clays redeposited into the Upper Pleistocene series was described for Tunel Wielki Cave[21,29]. Debris of layer 9 filled the deep vadose canyon cleaned by the previous erosion and achieved over 1 m thickness inside the depression, but it reached only limited thickness on the elevations, such as in the rear zone.

Layer 7 was deposited on the uneven surface of the rockfall debris and filled the depressions. The layer achieves the greatest thickness near the entrance and becomes thinner toward the cave interior. It disappears at a distance of approximately 300 cm from the dripline. It is a loess-like sediment or "cave loess"[27]. Respecting the lithostratigraphic scheme of those authors, the lower part of layer 7 (with limestone debris) correlates with their unit A, and its upper part (less debris, more silt) with unit C. Analogous sediments occur in a number of caves in the region, including the closest ones: Biśnik Cave and Shelter above the Zegar Cave [26,141]. Indistinct lower boundary with layer 9 indicates gradual change in sedimentary environment from physical disintegration of the bedrock related to cold and humid climate[6] to eolian accumulation typical for cold and dry conditions.

Intercalation of dark-colored layer 8 allows dividing layer 7 into sublayers 7a and 7b, but this possibility occurs on a limited area. Sublayer 8a in the lower part of layer 8 represents an anthropogenic activity. This deposit was formed by firing a hearth in a shallow pit dug in the loess. Few faunal remains have been found in layer 8. These include both the bones of larger mammals (hare and elk), which may be related to human hunting activity but also the remains of rodents and snails, likely deposited by natural agents. Presence of fauna and charcoals records a temporal climatic warming, replaced later again by periglacial conditions as marked by the overlying loess of layer 7a. Trough cross-stratification visible in sublayer 7a also indicates a role of colluvial processes in the formation of this stratum.

**Series IV–colluvial sediments.** Holocene sequence starts from layer 6, as indicated by radiocarbon dating (Table 1) and explosion of the abundance and diversity of faunal remains (Fig 11). Trough cross-stratification and lamination in layers 6 and 5 are distinct indicators of

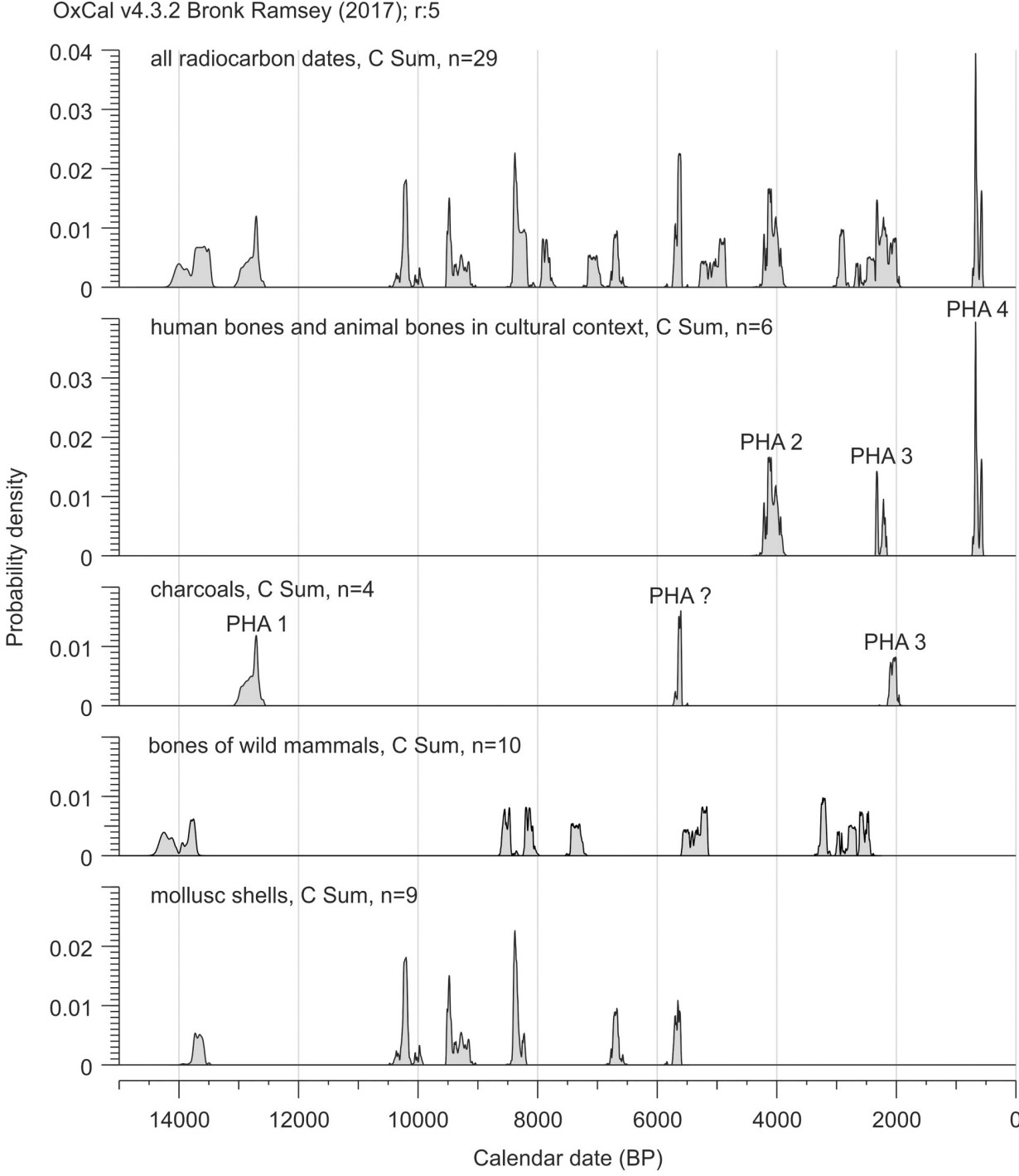

**Fig 11. Summary distribution of radiocarbon probabilities for ShSIII.** Calibrated dates are used. PHA are phases of human activity, described in the section "Chronometric data".

erosion and colluvial redeposition of sediments by washing and/or mudflows. Inclination of sedimentary structures indicates that the main direction of a geological transport was oriented from the rear zone of the rockshelter to the entrance and probably outside. Redeposited

material differs from any lower sediment by the higher amount of organic matter, Ca and P. This result means that before the redeposition, but after the sedimentation of layer 7, another type of sediment was accumulated in the rockshelter. The original strata did not survive the denudation, but their material has been preserved in a form of layers 6 and 5. The source of this material was most likely situated in the rear zone.

This series represents a period of geomorphological instability, when episodes of erosion and colluvial redeposition alternated several times in the rockshelter. The reason for these processes was likely high water supply, possibly linked with humid climate or degradation of the Pleistocene permafrost[3]. The series is topped with an unconformity, which may be regarded as the last record of erosional-colluvial activity. Records of analogous processes were recognized in other cave sites in the region[27] and linked with the lithostratigraphic unit E of the loess-like cave deposits.

**Series V–massive loams.** Massive (structureless) silts and loams with limestone debris of layers 5a, 4, 3a, 3 and 2 represent another series. No sedimentary macrostructures except for interlayer boundaries are visible, and even these boundaries are indistinct and gradual. This indicates a low sedimentation rate and slow or almost ceased geological transport. High amount of limestone clasts and high Ca content in the fine fraction support a hypothesis of an autochthonous source of the clastic material. A high concentration of P indicates intense zoogenic accumulation of excrements of cave dwelling animals and/or carcasses of their prey. The limestone clasts are smoothed and even rounded-like. Madeyska[29] linked the smoothing process with both mechanical erosion and chemical dissolution by dripwater and a generally humid and warm climate[63]. Layers 5a and 4 are particularly distinct due to their bright color and higher amount of clay. Increased content of clay and yellowish color indicate chemical weathering of limestone, i.e., the dissolution and removal of carbonates, accompanied by deposition of water-insoluble clay residuum enriched in Fe oxides. The bright color and higher amount of clay in cave sediments are regarded as an indicator of warm climate[56] and allow correlating of these layers with the Holocene climatic optimum.

In the entrance zone, the stratigraphy is less distinct due to pedogenesis (see the section "Pedogenesis" below). Noteworthy is the presence of large limestone blocks in layer 3. Some of them reach more than 90 cm in diameter. These blocks represent a massive rockfall event. Other large blocks (some of them longer than 1.5 m) protrude from the ground at the plateau in front of the rockshelter and possibly represent the same episode. As the excavation area uncovered only a limited part of this accumulation of blocks, it is difficult to conclude regarding its origin. It could represent the climatic conditions favoring physical weathering and disintegration of rock, i.e., a cold climate, but this is in contradiction with the record of a climatic optimum within the series. The tectonics can be excluded, as there are no signs of recent tectonic activity in the rockshelter, nor in the region. An alternative explanation is a catastrophic event, such as an accidental fall of a heavy block from the upper part of the hill, which could cause a massive avalanche.

**Series VI–dark humus-rich top sediments.** The uppermost layers 1 and 1a differ from sediments of the underlying series V. The most impressive feature is their dark color, related to increased humus content. The sand fraction is clearly higher. The unusually high concentrations of Pb and other toxic elements (Ag, Bi, Cd, Sb, Sn, Zn) suggest an increased impact of human activity. However, no archaeological artifacts made from these metals were found in the rockshelter. This suggests indirect pollution from the outside. It may be noteworthy that lead, zinc and silver mining in the southern part of Kraków-Częstochowa Upland has been active at least since Middle Ages[142] and is still ongoing. This medieval and modern industry could have polluted the region, e.g., via an eolian activity. Sediments of this series contain detrital material, i.e., bones with quite early radiocarbon dates (Fig 6). These bones likely come

from the backfill of archaeological feature, layer A, and testify later disturbance of the grave, possibly by burrowing animals.

## Postsedimentary disturbances

The sediments of ShSIII are relatively intact. Cryoturbation and bioturbation have had limited importance and affected only narrow stratigraphic horizons (Fig 10). The most extensive disturbances are those produced by pedogenesis and human activity.

**Cryoturbation.** Although the thick series of sediments were connected with cold climate (layers 7 and 9), the traces of cryoturbation are restricted in the sedimentary profile. The only distinct disturbances caused by frost action are visible in layers 10 and 11. A complex of these layers together with frost structures are cut by the unconformity and covered by layer 9. This indicates that the cryoturbation event took place before the deposition of layer 9 and the preceding erosion.

**Bioturbation.** Larger fossil burrows were found inside layers 5a and 5. The size of burrows suggests a constructor of a badger or fox size. These structures were filled with a sediment resembling layer 4. One of badger bones found inside one burrow gave a radiocarbon age of 5.0–4.8 ky BP (Poz-53303, Table 1). Some further dates obtained for layers 5a and 5 (Poz-61237 and Poz-61305) are consistent with dating of layer 4 (Table 1). These data allow correlating the burrowing activity with the deposition of layer 4 or a period soon after.

The presence of a brown bear skeleton[58] suggests that the rockshelter was used by bears as a hibernation den. It cannot be excluded that bears dug the hibernation pits and disturbed the sediments. However, the bones of a bear were found in a secondary position inside the backfill of an anthropogenic feature (layer A). The direct relationships between the bear remains and the sediments cannot be reconstructed.

An interesting example of the recent bioturbation are funnel-shape pits in the modern ground surface, dug in the layer 1a and filled with excrements, possibly of a badger. Two such structures were found during excavation, each approximately 20 cm deep, approximately 20–30 cm wide in the upper diameter and narrowing downward. The total volume of excavated excrements was approximately 10–12 liters. These pits can be interpreted as animal latrines. Production and use of such communal latrines is a known behavior of badgers[143,144]. Such badger activity might be responsible for the lateral scattering of some material from the backfill of the grave (layer A) throughout the rockshelter, and its incorporation into layer 1a.

**Anthropogenic disturbances.** The first sediment-disturbing human activity in the rockshelter was related to PHA 1 and happened during the Late Paleolithic. It affected the loess layers 7b and 8 in the area close to the rockshelter entrance. The Paleolithic visitors dug an oval pit that served as a fireplace. This trough-like object had a diameter of 60 cm and a depth of 30 cm and must have been filled with loess sediment (layer 8a) soon after humans had left the site.

The second distinctive phase of the anthropogenic disturbances is related to the burial of a woman in the late Middle Ages (PHA 4). The grave was constructed in the central zone and covered an area of approximately 2 m$^2$ (at squares D/5, D/6, E/5, E/6 and E/7). The length along NE-SW line is approximately 200 cm, whereas the width along NW-SE line is approximately 100 cm, and the greatest depth reaches 60 cm. The object contacts the limestone walls of the rockshelter (the bedrock) from the west and south-west, while several larger stones that can be interpreted as a relic of a purposefully built wall have been found in the entrance zone, at the north side of the grave[58]. The boundaries of the grave cut primarily layers 1, 1a, 2 and 3 and partially layers 4 and 5a in the southern part. The object is filled with sediments named layer A (field numeration "1a/2" and such name was used in a previous paper[57]). Inside the

entire backfill of the object, charcoal dust and fine charcoals were dispersed. Lithologically, the backfill resembles layers 1, 1a and 2, whose material was probably used to fill the grave. The backfill does not exhibit any visible stratification. This suggests a quick and single-event deposition of the sediment. Toward the bottom, the sediment becomes lighter in color and more clayey, similar to layer 4. This is probably a result of using the redeposited sediment of layer 4 during the initial filling. Archaeological material within the object is associated with different phases of human activity (PHA 2, PHA 3, and PHA 4, see Fig 11). It is mixed up and does not exhibit any systematic layout, except of human bones linked to PHA 4. This indicates that medieval people destroyed the earlier objects and used their backfills while constructing the grave. It seems probable that other features (perhaps graves or pits) associated with PHA 2 and PHA 3 were present in the central zone before the Middle Ages. However, the construction of the medieval grave completely consumed these older objects.

**Pedogenesis.** In the entrance zone, the stratigraphy of the upper part of the sequence is obscured by dark coloration, related to a high content of organic matter. The dark color becomes gradually lighter downward from the top of layer 1 to the bottom of layer 3, which is also marked by the gradual decrease in organic matter (Fig 5). This indicates that the organic matter was dispersed in profile by illuviation. Layer 1 represents an A-horizon (topsoil) and layer 3 represents a B-horizon (cambic) of a cambisol. The development of the soil was (and still is) possible here due to localization outside of the cave, as layers 1 and 3 occur only in the entrance zone, close to the dripline or outside of it. Recently, this area has been covered by forest litter composed mostly of beech, maple and oak leaves (Fig 2).

Layer 3a possibly represents a separate, earlier phase of pedogenesis. This stage of soil development was possibly halted by the erosion recorded in the bottom of series V and/or the accumulation of large limestone blocks visible now inside layer 3 (see the section "Series V–massive loams" and Fig 4). Some strata situated inside the rockshelter, namely, layers 1a and A and to a lesser extent layer 2, are also dark-colored and enriched in organic matter. This enrichment can be an effect of occasional input of litter to the interior of the rockshelter or redeposition of the humiferous sediment of layer 1 from the entrance zone (or topsoil material from outside the rockshelter) by wind, sheet wash and/or walking animals and humans.

## Age model

**Chronological frame of the sequence.** The radiocarbon and IRSL dates indicate the Late Pleistocene and Holocene chronology of the sediments in ShSIII. Dates arranged by layers show a consequent chronological order (Fig 6). The only inconsistency that prevented the OxCal sequence model from running (two dates were removed from the model; see the section "Chronometric data") is connected with layers 4-5a-5. It is noteworthy that these layers were a subject of bioturbation, as burrows filled with sediments of layer 4 were detected in layers 5a and 5 (see the section "Bioturbation"). That burrowing could be responsible for mixing of the subfossil material and observed chronological disorder. Two dates removed from the model (Poz-61237 and Poz-61305) were achieved on snail shells excavated from layers 5a and 5 but fit well to the set of dates for layer 4. These two dates likely represent intrusive material according to the terminology of Bronk Ramsey[74], in this case material from layer 4 that had been redeposited to the lower layers through animal burrows.

**Pleistocene sediments.** No dates are available for the lowermost layers 9, 10 and 11. The chronology of these layers can be approximated as being older than the earliest date accessible for the upper strata, which is 28.9 ± 1.7 ky BP (Table 3). Layers 7b and 8 clearly represent the Late Pleistocene. The radiocarbon dates fall between 14.5 and 12.5 cal ky BP for both layers. Bayesian modeling estimates the age of lower boundary of this series to be between 16.2 and

13.8 ky cal BP with 95.4% probability (see Table 3 and the section "Chronometric data"). The IRSL dates indicate much older age of sublayer 7b, approximately 28–21 ky cal BP. These dates likely represent a periglacial conditions of loess accumulation, when biogenic material was not deposited. Loess accumulation in the region, including its eolian deposition within the caves, has been dated to 27–11 ky BP[27]. No dates are available for layer 7a. The modeled statistical age of this layer is between 12.7 and 10.2 ky cal BP (with 95.4% probability; see Table 3 and the section "Chronometric data") and correlates with the Younger Dryas stadial and/or Preboreal phase of the Early Holocene[145].

**Holocene sediments.**   Radiocarbon dates indicate almost continuous accumulation of biogenic material through entire Holocene (Fig 11). The greatest gap in the probability distribution of radiocarbon age falls before 10.5 ky cal BP, i.e., in the beginning of Holocene.

Series IV (layers 5 and 6) includes the oldest Holocene sediments. The series was deposited during relatively short time interval according to radiocarbon dates, which lasted for several hundred up to a maximum of three thousand years (Table 3). The IRSL date achieved for layer 6 shows a much older age of clastic material than radiocarbon dating of animal remains (Table 2) and corresponds with the age of layer 7b. Regarding the colluvial origin of the series, this date can likely indicate a colluvial redeposition of the material of layer 7b, which has been incorporated into layer 6. A relatively early age is also indicated by the IRSL date for layer 5a. This age is earlier than any other date obtained for layer 7b and may possibly record another source of material (layer 9 or sediments occurring outside the rockshelter?). It is difficult to conclude on the sedimentary process responsible for accumulation of this material. No macroscopic traces of colluvial transport were identified. The important role of zoogenic deposition (see the section "Series V–massive loams") suggests that deposition of dust or mud by walking and trampling animals could be one of the accumulation processes.

The longest time interval is recorded by layer 4. Its deposition lasted for 4–5 thousand years according to the Bayesian model (Table 3). This interval corresponds to the late part of the Early Holocene and the entire Middle Holocene according to the formal stratigraphic subdivision of the Holocene[146]. It also corresponds to the entire Atlantic climatic chronozone[145]. Layer 3a from the entrance zone correlates with layer 4 according to the dating results and similarities in fossil rodent fauna, while the correlation between layers 3 and 4 is suggested by fossil rodent and mollusk fauna (see the sections "Malacological succession" and "Rodent succession").

Layer 2 corresponds with Late Holocene. Poor chronometric data are available for layers 1-1a. However, assuming that the archaeological feature corresponding to PHA 4 cuts these layers and the lower boundary of PHA 4 was estimated to be 2.0–0.7 ky cal BP (Table 3), the accumulation of layers 1-1a started before that phase. This dating is in good agreement with the modeled boundary of layers 2 and 1a, which is 2.9–2.4 ky cal BP (Table 3).

## Paleoecology and reconstruction of paleoenvironment

The abundance of fossil material makes the ShSIII an important paleontological site. Although Holocene paleofaunas were reported from other caves of southern Poland and adjacent countries, only few of these sequences go back as far as the final part of the Last Glaciation[11–16,41,43]. Most of the profiles are only a few dozens of centimeters thick and cover only the middle or just the youngest part of the Holocene[32–38]. Despite some disturbances noted in the narrow stratigraphic horizons in the ShSIII, the composition of the mollusk and rodent faunas appears to be environmentally coherent, which makes both groups good paleoenvironmental indicators.

**Reliability of paleoenvironmental reconstruction.** The mollusk thanatocoenoses in caves represent well the diversity of habitats in the surroundings, comprising shells of species penetrating the cave, of those living on the surface near the cave entrance and the shells dropped from the surrounding rocks during sedimentation[147,148]. The rather good state of preservation of shells in ShSIII suggests that they did not undergo any intensive transportation but rather were accumulated in sediments in the primary place of deposition after the mollusk death.

The rodent remains were deposited mostly from pellets dropped by carnivorous birds, such as owls resting in a cave[7]. Rodents likely could have inhabited a range of habitats surrounding the cave, which had been used by the birds as their hunting territories. Therefore, rodents represent a wider area than mollusks. It is also expected that predatory birds prefer open habitats over canopy for use as hunting grounds[149,150], so as a consequence, the abundance of open-habitat rodents can be overrepresented in the fossil record.

Bats are unique among the fossil fauna, as they are troglophiles who live inside caves. Some of the recorded taxa are known to hibernate in caves[151], while the others use caves as their summer roosts[95]. Bats are also migrants and some species move between their winter and summer areas at the distance of over hundred kilometers. Thus, species who used the caves as their hibernacula do not represent the habitats surrounding the cave. This may be the case for *M. dasycneme*. This species is known to hibernate in caves[151], so its presence in ShSIII is possibly an effect of death during winter. The water bodies used by this species as hunting grounds[125,126] were located in its summer area, which could be far from the site.

**Late pleistocene.** The oldest assemblages of mollusks and rodents (Vt and Dt, respectively) found in layers 7b-8-7a represent a rather poor fauna. Both assemblages are composed of species of cold climate, known from tundra and steppe biomes. Cold-tolerant snails *Vallonia tenuilabris* and *Discus ruderatus* indicate a cool continental climate and predominance of open habitats around the cave. The same habitats are preferred by lemmings. Dry open environments, i.e., steppe, are also indicated by the presence of *C. cricetus* and *S. subtilis*. However, some patches of cool, humid and shady habitats favorable for *S. kotulae*[83] also occurred, which indicates a mosaic of habitats. The rough and hilly topography of Kraków-Częstochowa Upland certainly was a factor favoriting the diversity of habitats. This is confirmed by regional paleobotanical data[152], which indicate that during Late Glacial of the Last Glaciation the clusters of trees dominated by pine, spruce and larch occurred beside the rocky slopes occupied by steppe-like communities.

The most informative is layer 8 with the highest number of shells. The occurrence of *S. kotulae* and *D. ruderatus* and lack of species typical of arctic climate, e.g., *Pupilla loessica*, *Vertigo genesii* and *Vertigo geyeri*, may indicate Bølling-Allerød interstadial[128,153]. Similar malacofaunas of this age are known from calcareous tufas and landslide deposits of southern Poland[34,37,153–155]. The rodent remains, though still low in number, follow the mollusk trend. They document higher abundance of woodland and humid environments during deposition of layer 8 in comparison to layers 7a and 7b. Additionally, the first appearance of a woodland species *C. glareolus* is recorded in layer 8.

**Pleistocene-holocene transition.** Rich and diverse mollusk fauna of DrVc assemblage points to the beginning of the Holocene and amelioration of the climatic conditions compared to the Late Glacial. Radiocarbon dating of layers 6 and 5 from the central zone and 3a from the entrance zone indicate the Preboreal and Boreal phases of Early Holocene, and possibly the very beginning of the Atlantic phase [145]. The mollusk composition resembles the *Ruderatus*-fauna[156] known across Europe from many terrestrial sequences between the Preboreal and the older part of the Atlantic phase[14,16,77,79,128–130,153,157,158,159–163]. Expansion of shade-loving taxa, namely, *Discus ruderatus*, suggests growth of taiga-type coniferous forests,

but some grasslands and sunny slopes preferred by *Vallonia costata* also occurred around the site. If shells have not been reworked, the environmental conditions in the beginning of Holocene favored the longer occurrence of glacial relics *Semilimax kotulae* and *Vallonia tenuilabris*. The last one was noted in Preboreal deposits in the nearby Cave above the Słupska Gate[41]. Gradual disappearance of glacial relics may correspond to gradual climate warming and some oceanic influences inferred from the first appearance of *Sphyradium doliolum*, *Discus rotundatus*, *Aegopinella pura* and other species of warm and humid conditions[41] (Fig 7).

The rodent assemblage CgLg found in layer 6 is still not abundant; however, it exhibits the highest Simpson index in the sequence. This reflects the high diversity of species and of habitats, as both the species of forest and of open landscape occur. Like in the case of snails, the rodent glacial relics survived, namely, *Dicrostonyx torquatus* and *Lasiopodomys gregalis*. According to Nadachowski[150], the glacial relics could survive in the region until the beginning of Late Holocene. Layer 6 is marked by the first appearance of *M. subterraneus* and genus *Apodemus* in the sequence. The first one was found in Late Glacial deposits of Shelter above the Niedostępna Cave[21,164], which is situated approximately 26 km away. In ShSIII this species is lacking in strata of that age and becomes abundant starting from layer 5 upward. This observation suggests reconsidering the early appearance of *M. subterraneus* in Shelter above the Niedostępna Cave, which could be an effect of postdepositional reworking. In another site with well-preserved Holocene deposits, Nad Mosurem Starym Duża Cave, its first appearance is even later, estimated to be ca. 4 ky BP[43].

Rodents of layer 5 record rapid improvement of environmental conditions. This is reflected by the increase of the remain number and higher amount of forest dwellers, such as *C. glareolus*, red squirrel *Sciurus vulgaris* and glirids *Eliomys quercinus* and *Muscardinus avellanarius*.

Interesting issue is a stratigraphic (and time) shift between the start of dominance of forest fauna in the mollusk and rodent records. The pick of forest-dwelling mollusks begins in layer 6 (or in layer 3a in the entrance zone), while in the case of rodents, only in layer 5 (or 3 in the entrance zone). This situation may be an effect of taphonomic biasing. As the accumulation of rodents prefers the taxa of open landscape (see the section "Reliability of paleoenvironmental reconstruction"), the dominance of forest rodents in the taphocenosis could have been intense since the forested area was virtually the only habitats in the surroundings, although patches of forest inhabited by forest rodents had been abundant earlier. The lack of chiropteran remains in the lower layers and their scarcity in layers 6 and 5 may be an effect of the unfavorable environmental conditions.

**Holocene climatic optimum.** Predominance of mixed and deciduous forests around the cave during sedimentation of layers 5a, 4 and 3 is evidenced by DroAp mollusk assemblage, CgMs rodent assemblage, PaurMbra chiropteran assemblage and radiocarbon dating (9.5–4.5 ky BP). *Discus ruderatus* was replaced by more demanding *D. rotundatus*, *D. perspectivus* and other strictly forest species, e.g., *A. pura*, *Balea biplicata*, *Ruthenica filograna* and *Isognomostoma isognomostomos* (Fig 7). This records the progressive warming and was likely connected with transition from continental to oceanic circulation across Central Europe and the climatic optimum of the Holocene–the Atlantic phase[77,129,130,159,163,165–167]. These data are in agreement with floristic changes in Central Europe[168,169]. The climatic optimum is indicated by the presence of *Vestia elata*, which has narrow ecological tolerance and favors warm and humid conditions. This snail has been described from Atlantic phase at other sites in the region and disappeared during the Subboreal phase[11,12,170]. Today, only limited relict sites with this species are noted in Poland in the Beskidy, Bieszczady and the Holy Cross Mountains[171].

The composition of rodent taphocenosis in layers 5a, 4 and 3 is similar to that in layer 5. All these layers are characterized by the highest richness of taxa (Fig 8). The important difference in comparison to layer 5 is much higher number of remains. Simpson index of diversity is

lower than in most of other layers, which is a result of strong predominance of two species, *C. glareolus* and *M. subterraneus*, both typical for woodlands.

The chiropteran assemblage PaurMbra, found in these layers, is also among of the richest in the sequence, both in terms of number of taxa and number of remains. The thanatocoenosis is characterized by taxa which prefer cold caves as their hibernacula, with temperature below 6˚C (which stays in accordance with small size of the ShSIII and testifies that the shelter already had than a similar size to the modern one), and big trees as summer roosts (*B. barbastellus*, *M. bechsteinii* and *P. auritus*). This suggests the presence of old-growth forest with sparse open areas (because of the presence of *Eptesicus serotinus*). The occurrence of *Rhinolophus hipposideros*, a species that typically prefers warmer caves to hibernate, could be limited to the summer period.

**Post-optimum deterioration of habitats.** Starting from approximately 5 ky BP, the malacofauna of ShSIII is dominated by mesophilous species. The forest malacofauna is still diverse but significantly outnumbered, mostly by *Laciniaria plicata* (Fig 7). The overall tendency of an increasing number of gastropods of wide ecological tolerance in costs of shade-loving taxa suggests a gradual deforestation and exposure of rocky walls around the cave. A similar situation is recorded in other Subboreal and Subatlantic sequences from caves and rockshelters of Central Europe[14,16,35,36,41,163].

The upper part of the sequence is characterized by a high abundance of edible dormouse *Glis glis*, which is the third most numerous rodent species in layers 2 and 1a. Its abundance probably coincided with a change in forest cover around the cave and rise in mature beech forest, which is a preferred habitat of this species[102,114]. Beech forest also constitutes the main type of the recent natural vegetation in the microregion. The expansion of this tree into the region was quite late, the beech forests appeared here approximately 3.5 ky BP[172]. This dating corresponds well with the presumed chronology of the mentioned layers.

The uppermost layers 1 and 1a record also the first appearance of *M. arvalis* and *Micromys minutus*. The arrival of both these species was most likely of synanthropic character, as they prefer artificial landscapes (agricultural fields and meadows). A similar observation of the appearance of *M. arvalis* during the late part of Late Holocene was noticed from other sites in the region[150]. Particularly, the Holocene profile of Nad Mosurem Starym Duża Cave shows the same pattern of rodent succession[43]. The presence of single remains of tundra species (*Dicrostonyx torquatus*) in layer 1a may be an effect of post-depositional disturbances.

Chiropteran fauna notes decline in richness and number of specimens, following the trend in mollusks and rodents. There is a clear predominance of taxa foraging in wooded areas (i.e., *B. barbastellus*, *M. bechsteinii*, *P. auratus* and *M. brandtii*). The presence of *M. myotis* can be related to a lack of undergrowth in the surrounding forest, while *M. mystacinus*, *M. nattereri*, *M. emarginatus* and *P. austriacus* suggest the presence of open and/or diversified areas, too. All these data likely indicate deterioration of habitats, most likely of anthropogenic origin.

## Chronology of human activity

Four episodes related to human activity have been recorded at ShSIII (see Fig 11). The trace of the oldest is the relic of a Late Paleolithic hearth (PHA 1), probably related to the short-term camp[59]. Another episode (PHA 2) is associated with a possible burial containing human and animal remains, which has been fragmentarily preserved in the secondary deposit inside the backfill of the medieval grave, and whose radiometric dating implies the activity of an Early Bronze Age community. The next phase of human stay (PHA 3) happened in the late pre-Roman period. Fragments of the vessel[58] and the animal deposit[60] were all redeposited into the medieval grave. Their context and nature imply the sepulchral character of human

activity from 4th to 1st century BC. Typology of the vessel based on stylistics and comparison with the other sites indicate the second half of the 1st century BC. Radiocarbon dating of the residue collected from the inner surface of the vessel fragment and of the wildcat mandible with cut marks indicate somehow earlier chronology, between the 4th and 1st centuries BC[58]. The youngest phase of human activity in the shelter (PHA 4) is a late Middle Age burial.

The cultural episodes identified at the rockshelter have numerous analogies in the nearest area. The Early Bronze Age sites are common in the caves of the region. The archaeological material (mainly pottery and flint artifacts associated with the production of tetrahedral axes) indicates that human activity in the other caves in the region was limited primarily to the plateaus outside of the caves or elevated places over the caves[45,141,173–176]. During the late pre-Roman period, the human activity in the caves of the Kraków-Częstochowa Upland increased. Ceramic vessels are the most common artifacts in other sites, but coins, elements of belts, fibulae and toiletry objects have also been found. Typically, the archaeological inventories suggest the economic and utilitarian meaning of caves for those people[177,178]. It should be emphasized that the nature of deposits in the ShSIII, i.e., a buried vessel and a sepulchral deposit of the European wildcat (*Felis silvestris*), make them unique among the Celtic sites in the region[58].

There are many examples of human activity in the caves of Kraków-Częstochowa Upland during the Middle Ages[9]. It is highly probable that the medieval burial in ShSIII could be related to the medieval castle Smoleń situated on a neighboring hill and the nearby village Pilica. However, the mysterious fact is a human burial inside a cave. This happened several centuries after the Christianization of the region, when cemeteries were commonly used as burial places.

Considering the small space of the rockshelter, the multicultural deposits and their sepulchral nature indicate the supermaterial significance of the place from Antiquity to the Middle Ages. It seems that the rockshelter has been visited for sacral or magical purposes[58].

## Shelter in Smoleń III as a lithostratigraphic stratotype

The almost two meters-thick sequence of Late Glacial and Holocene sediments in ShSIII is unique among the caves of Kraków-Częstochowa Upland and the caves of Central Europe in general. Although Upper Holocene cave sediments are quite common in the southern Poland (e.g., [32,42,43,179]), the Lower Holocene is preserved only on very rare occasions. In case of ShSIII, the entire profile of Lower-Middle-Upper Holocene is preserved, which is an exceptional situation. Taking this into consideration, we propose to regard ShSIII as a regional stratigraphic stratotype of Holocene clastic cave sediments. The synthetic stratigraphy of ShSIII is presented in Fig 12. The lithology and geochemistry of sediments allow reconstruction of the depositional conditions and can be regarded as a reference for future studies of cave sediments in the region. The main lithostratigraphic similarities shared by ShSIII with other sites in the region include:

1. Loess deposits in the lower part of the sequence (series III). Loess and loess-like sediments are widely known from the near-entrance facies of caves and rockshelters in Central Europe. A synthetic approach to their lithostratigraphy has been presented for Kraków-Częstochowa Upland[27], and lithostratigraphic units 'A', 'C' and 'E' of that scheme can be identified in ShSIII.

2. A dark-colored, humiferous stratum representing Bølling-Allerød interstadial (i.e., layer 8 here), which forms an intercalation inside a loess packet. Similar strata have been detected in Pekárna, Barová and Nová Drátenická caves in Czech Republic[14] and in Rockshelter in Zalas[32], however, in the latter the layer occurs within sands. Their age was confirmed by radiocarbon dating (16.2–14.2 in Pekárna Cave; 17.2–13.2 ky BP in Nová Drátenická Cave;

and 14.6–12.6 ky BP in Rockshelter in Zalas) and is similar to radiocarbon age of layer 8 in ShSIII.

3. Colluvial activity recorded in Lower Holocene (series IV). Traces of colluvial processes have been identified in similar stratigraphic position (at least, in the lower part of the Holocene series) in Cave above the Słupska Gate[41], in Zawalona Cave[13], in Rockshelter in Zalas[32], and in Nad Mosurem Starym Duża Cave[43].

4. Rusty-colored stratum with increased amount of clay in the Middle Holocene (here defined as series V). Recorded by layers 4 and 5a in ShSIII, this unit has analogies in Cave above the Słupska Gate[41], Rockshelter in Zalas[32] and in Pekarna Cave in Czech Republic[14].

5. Dark-colored humus-rich sediments at the top of a sequence (series VI). Similar strata are known from the most of caves and rockshelters in the region. However, they usually represent an entire Holocene sequence (e.g., [20–23,28–31]). The case of ShSIII shows that this situation is most likely an effect of removal of older Holocene sediments, followed by the most recent accumulation of series VI.

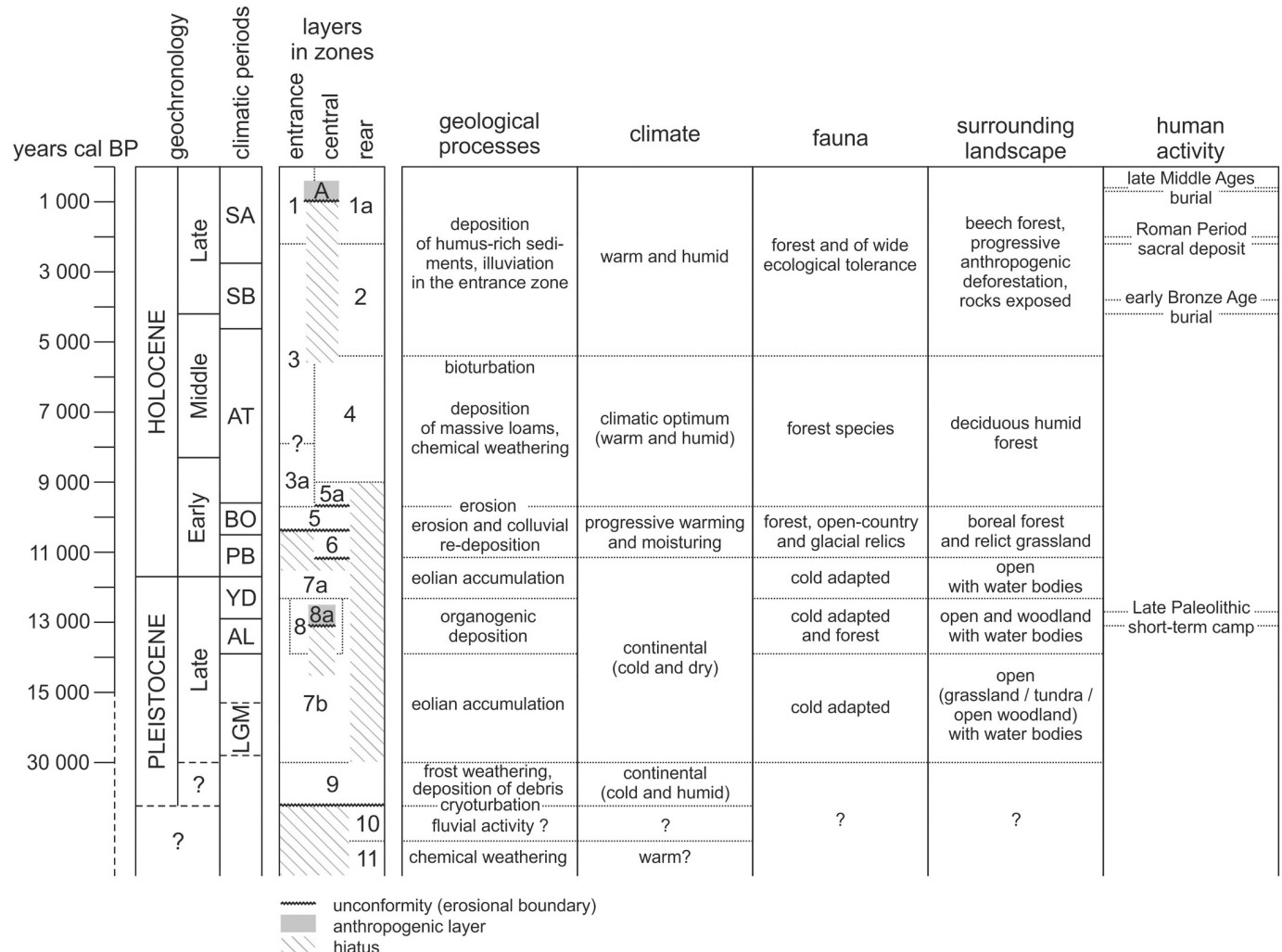

**Fig 12. Stratigraphy of the sediments from ShSIII and reconstruction of the paleoenvironmental conditions.** Boundaries of the geochronological units after Cohen et al.[146] and boundaries of the climatic zones after Starkel et al.[145].

The additional value of this site is that its stratified lithostratigraphic sequence contents an abundant mollusk shells and considerable number of vertebrate remains. Cave sites with long profiles of subfossil fauna of mollusks, rodents and chiropterans are unique for Central Europe and present a great value for reconstructions of paleoecology in the region[18]. The composition and structure of the mollusk and rodent assemblages and to lesser extent also chiropterans, allow reconstructing the changes in paleofauna, which reflected the changes of climate and vegetation around the cave. Several mid-European terrestrial sequences record changes in faunal succession of Pleistocene–Holocene transition and/or the entire Holocene (e.g., [14,16,17,41,128]). Among them, those with relatively rich paleontological material are in minority. Long and rich malacological successions are known from Barová Cave in Czech Republic[14], Veľká Drienčanská Cave in Slovakia[16] and Petény Cave in Hungary[17]. North to Carpathian Mountains, such records are known from Shelter over the Słupska Gate [41], Zawalona Cave[13] and Duża Cave at Mączna Skała[180]. Long and rich rodent successions are known from Barová Cave in Czech Republic[14], Nad Mosurem Starym Duża Cave [43], Rockshelter in Zalas[32] and Zawalona Cave[13]. Despite rather local character of the fauna diversity in those sequences, some general traits in transformation of fauna and their habitats may be distinguished throughout Central Europe (see, e.g., [35,36,79,148,150,152,163,168,169]). They show some common features with the sequence under study, such as:

1. Poor faunal assemblages of the Late Glacial of the Last Glaciation and Early Holocene.

2. Appearance of warm-adapted fauna in Preboreal and Boreal phases of the Early Holocene, recording the climatic warming and development of coniferous forests with patches of open habitats.

3. Deciduous and mixed forests of the climatic optimum, reflected especially in rich taphocenosis of snails and bats, with species of higher thermal and moisture requirements.

4. Deterioration of natural vegetation during the Late Holocene and increasing anthropopressure, visible in the appearance of species inhabiting open landscapes and synanthropic rodents. Increasing anthropopressure in the region is also marked by the presence of three human activity phases (PHA 2, PHA 3 and PHA 4) from Late Holocene, recorded in this small rockshelter in opposition to lack of any archaeological record from the Early or Middle Holocene.

## Conclusions

Almost two meters-thick sedimentary sequence of Late Glacial and Holocene sediments in ShSIII is unique among the mid-European caves, and among the caves of its northern part (Carpathian foreland) in particular. Although Holocene sequences are known from other sites in Central Europe, only on rare occasions they have been set in the certain chronological framework. Here we present a long sequence of clastic cave sediments which is also well-dated, with 29 radiocarbon and 4 IRSL dates provided and with Bayesian model of the chronology of stratigraphic units. This makes the ShSIII a key site for the understanding the regional Holocene lithostratigraphy. Based on the lithology and geochemistry of sediments we reconstruct the depositional conditions offering a reference for future studies of cave sediments in the region. We present also an abundant paleontological material, which records significant changes in paleofauna reflecting climate and environmental changes in the cave surroundings. The unusually long faunal succession places ShSIII among the crucial sites providing the

paleoenvironmental and paleoecological proxies for the Holocene of the Carpathian foreland. Taking all these facts into consideration, we propose to regard ShSIII as a regional stratigraphic stratotype of Holocene clastic cave sediments. The well-dated lithological sequence of ShSIII will serve as an important reference for future inter-site regional correlations. In particular, this creates an opportunity for archaeological regional studies, especially that archaeological cave sites with Holocene sediments are still excavated in Kraków-Częstochowa Upland[181–183].

Because ShSIII is a small rockshelter, the archaeological excavations consumed the greater part of its sediments (Fig 3). After the excavations, the profiles of unexcavated sections have been protected and the archaeological pits carefully backfilled with stones and previously sieved sediment. This makes the site secured and ready for eventual re-excavation in future, including sampling for further analyses. However, its unique scientific value makes it especially important to elaborate the project of legal protection for this site. The initial actions have already been made, as the surroundings of the site are included in the 'Natura 2000' protected area (*Ostoja Środkowojurajska*, site code PLH240009). Despite this, further particular protection of the cave is needed, especially in terms of the future scientific research and access to secured profiles.

## Supporting information

**S1 File. Lithological and geochemical parameters of sediments.**
(XLSX)

**S2 File. Mollusks–data on samples, habitats, number of specimens, individuals, assemblages and diversity.**
(XLSX)

**S3 File. Rodents–data on samples, habitats, number of specimens, individuals, assemblages and diversity.**
(XLSX)

**S4 File. Bats–data on samples, number of specimens, individuals, assemblages and diversity.**
(XLSX)

**S5 File. Archaeology–data on archaeological artifacts and anthropological remains.**
(XLSX)

## Acknowledgments

We are thankful to the officers of the Polish National Forests for their help with the organization of excavations; and to the students of archaeology from Nicolaus Copernicus University in Toruń who participated in the excavation works.

## Author Contributions

**Conceptualization:** Maciej T. Krajcarz, Marcin Szymanek, Magdalena Krajcarz.

**Formal analysis:** Maciej T. Krajcarz, Marcin Szymanek, Magdalena Krajcarz, Andrea Pereswiet-Soltan.

**Funding acquisition:** Maciej T. Krajcarz, Marcin Szymanek, Magdalena Krajcarz, Andrea Pereswiet-Soltan, Magdalena Sudoł-Procyk.

**Investigation:** Maciej T. Krajcarz, Marcin Szymanek, Magdalena Krajcarz, Andrea Pereswiet-Soltan, Witold P. Alexandrowicz, Magdalena Sudoł-Procyk.

**Project administration:** Maciej T. Krajcarz, Marcin Szymanek, Magdalena Krajcarz, Andrea Pereswiet-Soltan, Magdalena Sudoł-Procyk.

**Resources:** Maciej T. Krajcarz, Magdalena Krajcarz, Magdalena Sudoł-Procyk.

**Visualization:** Maciej T. Krajcarz, Marcin Szymanek.

**Writing – original draft:** Maciej T. Krajcarz, Marcin Szymanek, Magdalena Krajcarz, Andrea Pereswiet-Soltan, Magdalena Sudoł-Procyk.

**Writing – review & editing:** Maciej T. Krajcarz, Marcin Szymanek, Magdalena Krajcarz, Andrea Pereswiet-Soltan, Witold P. Alexandrowicz, Magdalena Sudoł-Procyk.

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
