## [Decision Letter · Decision Letter 0]

19 Nov 2019

PONE-D-19-27803

Shelter in Smoleń III – a unique example of stratified Holocene clastic cave sediments in Central Europe, a lithostratigraphic stratotype and a record of regional paleoecology

PLOS ONE

Dear Dr. Krajcarz,

Thank you for submitting your manuscript to PLOS ONE. After careful consideration, we feel that it has merit but does not fully meet PLOS ONE’s publication criteria as it currently stands. Therefore, we invite you to submit a revised version of the manuscript that addresses the points raised during the review process.

While reviewer 1 entered a 'reject' recommendation, this was only because they felt unqualified to examine the bulk of your manuscript. However, they found your study well-written and thorough, an assessment backed up by Reviewer 2's more extensive analysis. Nevertheless, both reviewers indicated that your manuscript lacked some important contextual information that would make it more relevant to general readers. In particular, mention is made of the importance of this site for archaeological and palaeoenvironmental research, but these areas are only minimally addressed in the discussion. They should be more thoroughly introduced at the beginning, with an effort made in the conclusions to tie the major findings of this research within this context. 

We would appreciate receiving your revised manuscript by Jan 03 2020 11:59PM. To enhance the reproducibility of your results, we recommend that if applicable you deposit your laboratory protocols in protocols.io, where a protocol can be assigned its own identifier (DOI) such that it can be cited independently in the future. For instructions see: http://journals.plos.org/plosone/s/submission-guidelines#loc-laboratory-protocols

We look forward to receiving your revised manuscript.

Kind regards,

Julien Louys

Academic Editor

PLOS ONE

Journal Requirements:

2. In your manuscript, please ensure that you have reported specimen numbers and complete repository information, including museum name and geographic location.

For more information on PLOS ONE's requirements for paleontology and archaeology research, see https://journals.plos.org/plosone/s/submission-guidelines#loc-paleontology-and-archaeology-research.

a) Please provide an amended Funding Statement that declares *all* the funding or sources of support received during this specific study (whether external or internal to your organization) as detailed online in our guide for authors at http://journals.plos.org/plosone/s/submit-now.  

b) Please state what role the funders took in the study.  If any authors received a salary from any of your funders, please state which authors and which funder. If the funders had no role, please state: "The funders had no role in study design, data collection and analysis, decision to publish, or preparation of the manuscript."

4. We note that Figures 1 and 3 in your submission contain [map/satellite] images which may be copyrighted. All PLOS content is published under the Creative Commons Attribution License (CC BY 4.0), which means that the manuscript, images, and Supporting Information files will be freely available online, and any third party is permitted to access, download, copy, distribute, and use these materials in any way, even commercially, with proper attribution. For these reasons, we cannot publish previously copyrighted maps or satellite images created using proprietary data, such as Google software (Google Maps, Street View, and Earth). For more information, see our copyright guidelines: http://journals.plos.org/plosone/s/licenses-and-copyright.

1. You may seek permission from the original copyright holder of Figures 1 and 3 to publish the content specifically under the CC BY 4.0 license. 

Reviewers' comments:

Reviewer's Responses to Questions

**Comments to the Author**

1. Is the manuscript technically sound, and do the data support the conclusions?

Reviewer #1: Partly

Reviewer #2: Yes

2. Has the statistical analysis been performed appropriately and rigorously? 

Reviewer #1: I Don't Know

Reviewer #2: Yes

3. Have the authors made all data underlying the findings in their manuscript fully available?

Reviewer #1: Yes

Reviewer #2: Yes

4. Is the manuscript presented in an intelligible fashion and written in standard English?

Reviewer #1: Yes

Reviewer #2: No

5. Review Comments to the Author

Reviewer #1: The site seems to be an important one for the thickness of its Late Quaternary deposits, quality and diversity of environmental proxy data, large number of good C14 and other radiometric dates, etc. It is well written (with only minor errors of colloquial English), excellently illustrated and quite adequately supported. But, as an archeologist of the Pleistocene and earliest Holocene, I was disappointed at the virtual absence of archeological (cultural) information or analysis. Nor am I a Central European prehistory specialist. So I am not an appropriate reviewer. My choice by PLOS-1 was not correct.

However, I cannot help but notice that this ms. is quite narrowly descriptive; neither the statement of purpose nor the conclusions really tell the reader why this site is important in the overall context of southern Poland or Central Europe generally. What does the site tell us about human activity, adaptations or behavior (especially in a broader context). I imagine that anthropological interpretation was not the authors' goal, but it is a shame that such an interesting site is not more generally contextualized in terms of humans and their cultures in this region throughout the course of the early Holocene. The Editors should seek appropriate reviewers who are relevant natural scientists.

Reviewer #2: The manuscript entitled Shelter in Smoleń III – a unique example of stratified Holocene clastic cave sediments in Central Europe, a lithostratigraphic stratotype and a record of regional paleoecology, deals with the stratigraphy of the archeologically valuable rock shelter. Sediment deposits have been analyzed in detail (morphological classification of limestone clasts, chemical composition...). 29 radiocarbon dating of faunal and human remains and charcoals cover the interval from 14.1 to 0.6 ky BP. Taxonomic composition of fossil faunal assemblages was used to obtain insight to the reconstruction of local paleoenvironment.

I would like to compliment the authors for their great effort collecting and analyzing such an important amount of data and effectuating a valuable multidisciplinary research. In cave entrances and especially in rock shelters, where sedimentation rates are high and artifact preservation can be remarkable, many aspects and processes must be considered, and the authors managed that requests very well.

The title is appropriate for the content of the text; hence the abstract could give more insight to the findings. Furthermore, the article is well constructed, the sampling and pitting were properly conducted, and analyses used are appropriate and well performed. Methodology and results are well described and accounts for large part of the manuscript. However, there are several issues that should be considered before publishing.

- The manuscript is clearly written. Still, several sentences are confusing in structure or word use. I suggested some changes but it would be good to edited the text by a native English speaker.

- Introduction should give better overview on the topic and the current state of the art in the field.

- The results are described in detail and are well interpreted, but a bit hard to follow.

- Discussion is structured in several sections, it is well explained but there is a clear need of more synthetic interpretations. All sections could be better linked together through time, discussed climatic events, paleo environmental changes and anthropogenic activities.

- Some statements should be more clearly explained and supported by references (for details please see the manuscript)

- I suggest moving the fig. 12 from Conclusion to Discussion, even to separate (sub)section which could be built around it.

- The manuscript would benefit putting the findings in broader context – regional settings (Central Europe, differences and similarities) with previous research.

- If the authors accept this suggestions Conclusion than could be restructured, giving more value to the authors results.

Overall, the research itself brings new paleoenvironmental data from a complex climate region, such as Central Europe, and contributes to better understanding of local climate dynamics and anthropogenic activities during the Holocene. However, more synthetic views would substantially increase the value of the presented results and consequently manuscript.

Additional comments can be found in the manuscript.

6. PLOS authors have the option to publish the peer review history of their article (what does this mean?). If published, this will include your full peer review and any attached files.

Reviewer #1: No

Reviewer #2: No

---

## [Author Response · Author response to Decision Letter 0]

14 Jan 2020

Response to reviews

We are very thankful to Reviewers for their careful review and many constructive comments. We added the corrections to text according to these suggestions. We also found some minor mistakes (typos) on some figures, and now we submitted the modified versions.

Below, we provide our detail responses to Reviewers and explain our corrections. The manuscript with our changes in a review mode is also attached as a separate file.

Reviewer #1: 

COMMENT: The site seems to be an important one for the thickness of its Late Quaternary deposits, quality and diversity of environmental proxy data, large number of good C14 and other radiometric dates, etc. It is well written (with only minor errors of colloquial English), excellently illustrated and quite adequately supported. But, as an archeologist of the Pleistocene and earliest Holocene, I was disappointed at the virtual absence of archeological (cultural) information or analysis. Nor am I a Central European prehistory specialist. So I am not an appropriate reviewer. My choice by PLOS-1 was not correct. However, I cannot help but notice that this ms. is quite narrowly descriptive; neither the statement of purpose nor the conclusions really tell the reader why this site is important in the overall context of southern Poland or Central Europe generally. What does the site tell us about human activity, adaptations or behavior (especially in a broader context). I imagine that anthropological interpretation was not the authors” goal, but it is a shame that such an interesting site is not more generally contextualized in terms of humans and their cultures in this region throughout the course of the early Holocene. The Editors should seek appropriate reviewers who are relevant natural scientists.

ANSWER: We agree that the site has a potential for archaeological interpretations. However, our purpose was to provide in this paper the environmental data (paleoecology and lithostratigraphy), which can be used in future for more detail inter-site correlations. This is also our plan for the future to present such correlations for the entire region of Kraków-Częstochowa Upland, and further, to set archaeological data within this stratigraphic context. Only then we would be able to present a coherent and reliable history of human activity in the area. And that’s our plan, but it needs several steps before, and this paper is thought to be one of them. 

Now, also to address to the Reviewer’s #2 comments, we have added new sections to the Introduction and Discussion and re-built the Conclusions, to provide a broader context of our study.

Reviewer #2: 

COMMENT: The manuscript entitled Shelter in Smoleń III – a unique example of stratified Holocene clastic cave sediments in Central Europe, a lithostratigraphic stratotype and a record of regional paleoecology, deals with the stratigraphy of the archeologically valuable rock shelter. Sediment deposits have been analyzed in detail (morphological classification of limestone clasts, chemical composition...). 29 radiocarbon dating of faunal and human remains and charcoals cover the interval from 14.1 to 0.6 ky BP. Taxonomic composition of fossil faunal assemblages was used to obtain insight to the reconstruction of local paleoenvironment.

I would like to compliment the authors for their great effort collecting and analyzing such an important amount of data and effectuating a valuable multidisciplinary research. In cave entrances and especially in rock shelters, where sedimentation rates are high and artifact preservation can be remarkable, many aspects and processes must be considered, and the authors managed that requests very well. The title is appropriate for the content of the text; hence the abstract could give more insight to the findings. Furthermore, the article is well constructed, the sampling and pitting were properly conducted, and analyses used are appropriate and well performed. Methodology and results are well described and accounts for large part of the manuscript. 

However, there are several issues that should be considered before publishing.

- The manuscript is clearly written. Still, several sentences are confusing in structure or word use. I suggested some changes but it would be good to edited the text by a native English speaker.

ANSWER: We are thankful for this advice. However, our manuscript has been sent before the submission to the company (American Journal Experts) which makes professional corrections of scientific texts, and has been extensively corrected by their native speakers who also are the experts in the field of this paper. 

COMMENT: - Introduction should give better overview on the topic and the current state of the art in the field.

ANSWER: Now we have provided a wider background according to this comment.

COMMENT: - The results are described in detail and are well interpreted, but a bit hard to follow.

ANSWER: Nevertheless, we would like to keep the current structure of the paper. It is quite long, so we believe it will be much easier for readers who are focused only on particular area of interest to find the particular section (for example, data on fossil rodents) if structured like it is now. 

COMMENT: - Discussion is structured in several sections, it is well explained but there is a clear need of more synthetic interpretations. All sections could be better linked together through time, discussed climatic events, paleo environmental changes and anthropogenic activities.

ANSWER: Now we have added new section to the Discussion to provide more synthetic approach. This also fills the requirement from the next comments.

COMMENT: - I suggest moving the fig. 12 from Conclusion to Discussion, even to separate (sub)section which could be built around it.

ANSWER: We agree with this advice. So now, we rebuilt the Discussion using parts of former Conclusion (including Fig. 12) and this gave also the linkage between different sections of Discussion and we believe this provided more synthetic approach. 

COMMENT: - The manuscript would benefit putting the findings in broader context –; regional settings (Central Europe, differences and similarities) with previous research.

ANSWER: Now our new sections in Introduction and in Discussion address to this constructive suggestion.

COMMENT: - If the authors accept this suggestions Conclusion than could be restructured, giving more value to the authors results.

ANSWER: Now we have re-built Discussion and Conclusions according to the comments.

COMMENT: Overall, the research itself brings new paleoenvironmental data from a complex climate region, such as Central Europe, and contributes to better understanding of local climate dynamics and anthropogenic activities during the Holocene. However, more synthetic views would substantially increase the value of the presented results and consequently manuscript.

- Some statements should be more clearly explained and supported by references (for details please see the manuscript)

Done

COMMENT: Additional comments can be found in the manuscript.

ANSWER: We corrected manuscript according to the advices provided by Reviewer in the attached PDF.

According to the Reviewer’s #2 questions: 

- line 867 – here we would like to say generally about the process, without referencing to chronological periods

- line 883 – there are no signs of neotectonics in the rockshelter nor in the area. We inserted the appropriate information into the text. 

- line 1069 – we did not identified fossil guano. However, the micromorphological analyses has not been implemented yet into this site, which in future can help to resolve this problem.

---

## [Editor Report · Decision Letter 1]

21 Jan 2020

Shelter in Smoleń III – a unique example of stratified Holocene clastic cave sediments in Central Europe, a lithostratigraphic stratotype and a record of regional paleoecology

PONE-D-19-27803R1

Dear Dr. Krajcarz,

We are pleased to inform you that your manuscript has been judged scientifically suitable for publication and will be formally accepted for publication once it complies with all outstanding technical requirements.

With kind regards,

Julien Louys

Academic Editor

PLOS ONE
---

## [Editor Report · Acceptance letter]

31 Jan 2020

PONE-D-19-27803R1 

Shelter in Smoleń III – a unique example of stratified Holocene clastic cave sediments in Central Europe, a lithostratigraphic stratotype and a record of regional paleoecology 

Dear Dr. Krajcarz:

I am pleased to inform you that your manuscript has been deemed suitable for publication in PLOS ONE. Congratulations! Your manuscript is now with our production department. 

With kind regards,

on behalf of

Dr. Julien Louys 

Academic Editor

PLOS ONE